# Impact of COVID-19 on antenatal care provision at public hospitals in the Sidama region, Ethiopia: A mixed methods study

Zemenu Yohannes Kassa[1,2]*, Vanessa Scarf[1], Sabera Turkmani[1], Deborah Fox[1]

**1** Collaborative of Midwifery, Child and Family Health, Faculty of Health, University of Technology Sydney, Ultimo, NSW, Australia, **2** College of Medicine and Health Sciences, Hawassa University, Hawassa, Ethiopia

* ZemenuYohannes.Kassa@student.uts.edu.au, zemenu2013@gmail.com

## Abstract

### Background

Coronavirus disease 2019 (COVID-19) continues to pose a global public health threat. The pandemic overstretched already weak health systems in low- and low-middle-income countries, including Ethiopia. There is a paucity of studies on the impact of COVID-19 on antenatal care access, uptake, and provision in Ethiopia. This study examines the impact of COVID-19 on antenatal care provision in the Sidama region, Ethiopia.

### Methods

A concurrent mixed-methods study was conducted between 14 February and 10 May 2022 at 15 public hospitals in the Sidama region. An interrupted times series design was applied for a quantitative study, which included data from all pregnant women who attended antenatal care before COVID-19 (12 months, March 2019 to February 2020) and during COVID-19 (six months, March to August 2020) at 15 public hospitals in the region. The total numbers in the antenatal care 1 cohort (at least one antenatal care contact) and antenatal care 4 cohort (at least four antenatal care contacts) were 15,150 and 5,850, respectively, forming a combined final dataset of 21,000 women. Routinely collected monthly data were derived from the hospitals' health management information system and imported into Stata version 17 for analysis. The mean monthly incidence rate ratio of antenatal care uptake was calculated using a Poisson regression model with a 95% confidence interval. Simultaneously, an exploratory study design was conducted for qualitative using in-depth interviews to explore maternity care providers' perceptions of the impact of COVID-19 on antenatal care access, uptake, and provision. Qualitative data were thematically analysed. The quantitative and qualitative findings were then integrated using the joint display technique.

### Results

Our findings indicate a significant monthly decrease of 0.7% in antenatal care 1 and 1.8% in antenatal care 4 during the first six months of the pandemic. A lack of medical supplies, fear of contracting COVID-19, inadequate personal protective equipment, discrimination against

**Data Availability Statement:** Data Availability Statement: The interview data cannot be shared publicly since it contains potentially attributable sensitive information regarding participants and

their roles. Sharing such data would violate and undermine the ethical committee agreement and consent process. Researchers who meet the criteria for access to confidential data may request it from the University of Technology Sydney Human Research Ethics Committee at Research. Ethics@uts.edu.au. All other relevant data are presented within the paper.

**Funding:** The Royal Society of Tropical Medicine and Hygiene is supported financially for only data collection. The funders had no role in study design, data collection and analysis, decision to publish, or preparation of the manuscript.

**Competing interests:** The authors have declared that no competing interests exist.

**Abbreviations:** ANC, Antenatal care; ARIMA, Autoregressive integrated moving average; CEmOC, Comprehensive emergency obstetric care; COVID-19, Coronavirus disease 2019; HMIS, Health management information system; IDI, In-depth interview; IESO, Integrated emergency surgical officer; IRR, Incidence rate ratio; ITS, Interrupted the time series; NICU, Neonatal intensive care unit; PPE, Personal protective equipment; SDG, Sustainable development goal; WHO, World Health Organization.

those attending the hospital, and the absence of antenatal care guidelines for care provision, COVID-19 vaccine hesitancy and long waiting times for ANC led to disrupted access, uptake, and provision of antenatal care during COVID-19.

## Conclusion and recommendations

Our findings demonstrate that the COVID-19 pandemic affected antenatal care access, uptake, and provision in the study area from March to August 2020. To mitigate disrupted antenatal care access, uptake and provision, antenatal care clinics should be equipped with medical supplies. It is crucial to maintain rapport between the community and maternity care providers and provide training for maternity care providers regarding the adapted/adopted guidelines during COVID-19 at the hospital grassroots level for use in the current and future pandemics. Pregnant women should have timely access to maternity care providers in order to maintain at least a minimum standard of care in current and future pandemics.

## Introduction

The coronavirus disease 19 (COVID-19) has created an unprecedented global public health crisis and continues to pose a global health threat [1]. It is a highly contagious viral pneumonia that causes severe acute respiratory syndrome (SARS-CoV-2) [2]. COVID-19 is a rapidly spreading virus with cases throughout the world since its first identification in Wuhan, China, in December 2019 [3]. The World Health Organization (WHO) declared COVID-19 as a public health emergency and pandemic in March 2020 [4].

Despite the range of prevention strategies employed to contain the COVID-19 pandemic, over 771.5 million people have contracted the virus, and as of 25 October 2023, more than 6.97 million related deaths had been reported globally [5]. More than 9.55 million COVID-19 cases in Africa and more than 175,443 deaths had been reported as of 25 October 2023 [5]. In Ethiopia, at that date, 501,060 COVID-19 cases and 7,574 deaths related to COVID-19 had been reported [5].

Over the past two decades, improvements in access to antenatal care (ANC) have substantially reduced maternal and neonatal morbidity and mortality in resource-constrained countries [6, 7]. ANC optimises maternal and child health outcomes through regular pregnancy monitoring [8], and provides an opportunity to offer care to prevent and manage existing health complaints and potential causes of maternal and neonatal illness [9]. However, the COVID-19 pandemic may have reversed improvements in ANC utilisation in low- and middle-income countries (LMICs). The pandemic also presents a profound obstacle to implementing the recommended guidelines for improving ANC health service utilisation [9].

In 2016, the WHO recommended a minimum of eight ANC contacts for all pregnant women [10]. Although ANC utilisation had increased during the two decades preceding the pandemic [11, 12], many countries had still not reached the WHO-recommended eight ANC contacts, and women in Ethiopia had fewer ANC contacts than in most other countries. Globally, and in Eastern and Southern Africa, 87% of pregnant women received at least one ANC contact (termed 'ANC1'), compared to only 74% in Ethiopia. In 2019, the percentage of pregnant women that received at least four ANC contacts (termed 'ANC4') was 66% globally, 54% in Eastern and Southern Africa [13], and just 43% in Ethiopia [14].

Since the beginning of the COVID-19 pandemic, ANC utilisation has significantly decreased in (LMICs) [15–18] due to fear of contracting the virus [15, 19, 20], lockdowns [21, 22], closure of health institutions [23], lack of transport [24], community fear of health institutions, and delay in healthcare-seeking [21, 25]. Consequently, stillbirths, maternal depression, and maternal deaths have all increased [26].

In the early stage of the COVID-19 pandemic, Ethiopian government, and nongovernmental organisations (NGOs) shifted their focus towards containing the spread of the virus by implementing a range of measures. These measures included declaring a state of emergency, reducing the passenger capacity in public transport by half, imposing a lockdown and encouraging people to stay at home [27]. Consequently, the lockdown measures resulted in job losses for many women [28]. These factors posed significant challenges to women's ability to meet their basic needs [28]. As a result, the provision of ANC has been and continues to be, impacted by the direct and indirect consequences of COVID-19 [29]. However, the existing studies in Ethiopia on the impact of the COVID-19 pandemic have not rigorously explored its impact on maternity care services, specifically ANC access, uptake, and provision. The paucity of studies on the impact of COVID-19 on ANC access, uptake and provision in Ethiopia made it essential to conduct this study to estimate and explore the impact of COVID-19 on the country's ANC access, uptake, and provision.

## Methods

This study is part of a larger mixed-methods investigation of the impact of COVID-19 on maternal and perinatal care at 15 public hospitals in the Sidama region of southern Ethiopia, carried out between 14 February and 10 May 2022. Sidama is the 10[th] newly established region in Ethiopia. The region is currently divided into 5 city administrations and 31 administrative divisions, known as 'Woredas'. In 2019/2020, the region's total population was 3,983,969, with 1,974,455 males and 2,009,514 females. Sidama has 928,265 women of reproductive age (15–49), 137,845 of whom gave birth in 2019/2020. Regarding healthcare provision, the region has 14 primary hospitals, 3 general hospitals, 1 comprehensive specialised teaching hospital, 123 public health centres, 526 health posts, and over 108 private clinics. Comprehensive emergency obstetric care was available in 15 public hospitals in Sidama during 2020.

### Study design and data source

We used a concurrent mixed-methods design [30] incorporating routinely collected quantitative data derived from the health management information system (HMIS) in hospitals [31]. An interrupted time series design was applied for quantitative and an exploratory design for qualitative study. Quantitative and qualitative data were collected in parallel, and the first stage of analysis separately addressed the two types. A further integration phase was conducted to compare and contrast the quantitative and qualitative findings and assess whether they were corroborative or contradictory by making mixed-methods meta-inferences. The integration was conducted by joint display technique [30, 32] (*Fig 1*).

### Data collection methods for quantitative data

Data were collected from all 15 public hospitals providing comprehensive emergency obstetric care in the Sidama region at the time of the study. The sample included all pregnant women who attended ANC in the 12 months before the COVID-19 pandemic (March 2019 to February 2020) and during the six months of the pandemic (March to August 2020), totalling 21,000 women. The first author and research assistants (all with an MSc in clinical midwifery) extracted data from the hospitals' HMIS. The first case of COVID-19 was officially reported in

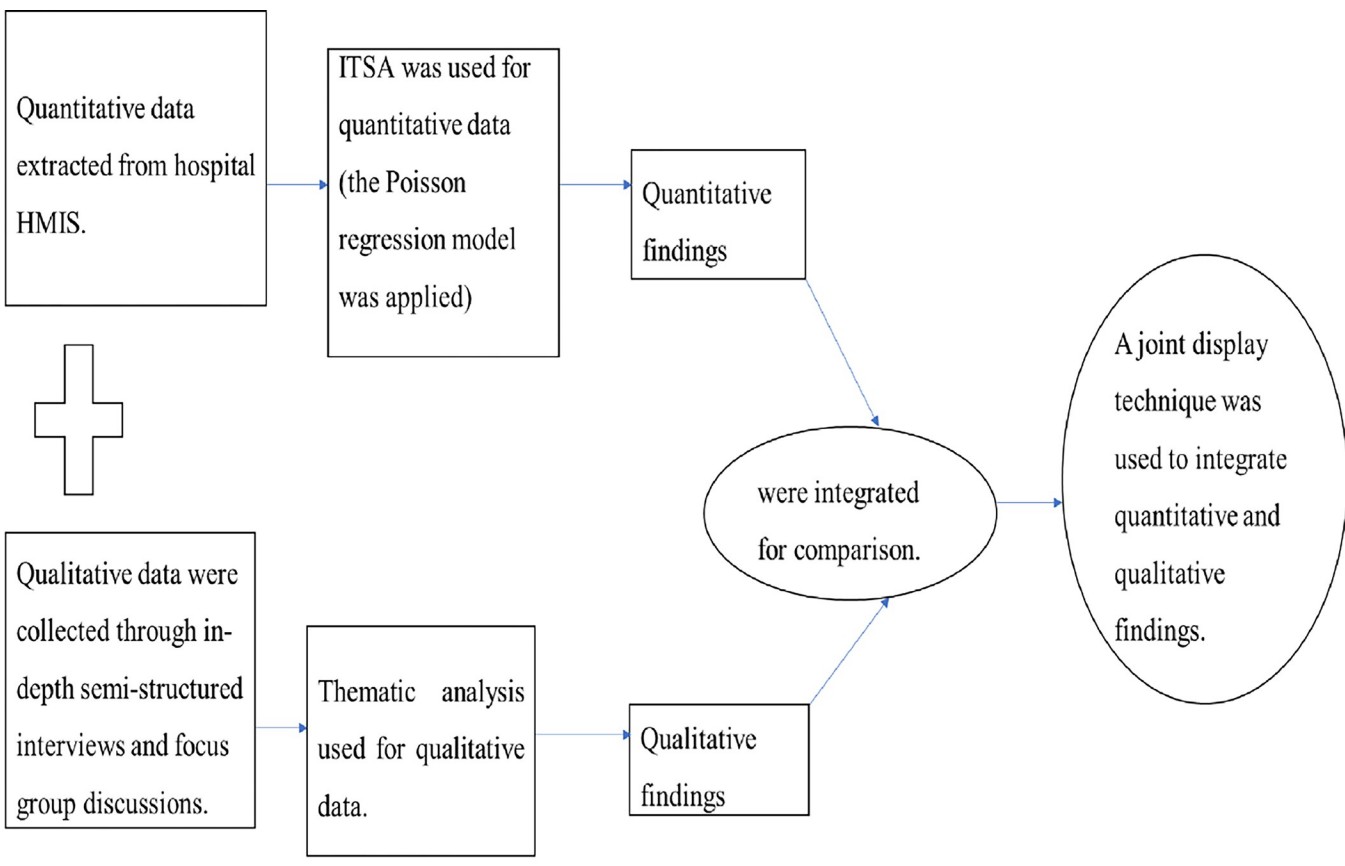

**Fig 1. Concurrent mixed methods research on the impact of COVID-19 on ANC provision was adapted [30].**

Ethiopia on 13 March 2020. We used the same months before and during the pandemic to assess the impact of COVID-19 on ANC uptake. Similarly, we included data from March 2019 to February 2020 to evaluate whether ANC was affected by other incidents before COVID-19 and whether these events affected ANC outcomes from March to August 2020.

## Quantitative data processing and analysis

After screening the data, any questions regarding data clarity were resolved by revisiting the hospitals and HMIS offices to address any missing data in the provided Excel spreadsheet. Data were imported from Microsoft Excel into Stata version 17 for analysis. We performed an interrupted time series analysis (ITSA) to estimate trends in the uptake of ANC across two periods: before COVID-19 (March 2019 to February 2020) and during COVID-19 (March to August 2020). ITSA can evaluate the impact of population-level interventions, including policy changes and infection prevention programmes, implemented at a clearly defined time [33]. The first official COVID-19 case in Ethiopia was reported on 13 March 2020, so we used this date to mark the start of the intervention period. The mean monthly incidence rate ratio (IRR) of ANC uptake was calculated with a 95% confidence interval (CI), using a Poisson regression model [34, 35] with pre-COVID-19 data as the reference. A Poisson regression model was suitable because the monthly reports of ANC provision comprised count data (non-negative integer values). In ITSA, a Poisson regression model [36] performs better than an autoregressive

integrated moving average (ARIMA) model, which is more conventionally used for real-valued time series data. Differences are considered statistically significant at a p-value of less than 0.05 ($p < 0.05$).

We used a single-group ITSA for this study [35]

$$Y_t = \beta_0 + \beta_1 T_t + \beta_2 X_t + \beta_3 X_t T_t + \beta_m month + \epsilon_t$$

where $Y_t$ is the aggregated outcome; $\beta_0$ estimates the ANC uptake number before COVID-19; $\beta_1$ estimates the average monthly change in ANC uptake before COVID-19; $T_t$ is the time since the start of the study; $\beta_2$ represents the change in ANC uptake occurring immediately during the pandemic (within three months) (designated by $X_t T_t$); $\beta_3$ denotes the difference between the trends in ANC uptake before and during COVID-19; and $\beta_m$ represents the month and $\epsilon_t$ the random error.

In this model, time is measured as a dummy variable, taking the value 0 for the period before COVID-19 and 1 for the period during COVID-19 (the intervention period) [35, 37].

## Study approach for qualitative data

We adopted an exploratory design [38] to investigate maternity care providers' views on and experiences of the impact of COVID-19 on ANC provision in the Sidama region. In-depth interviews (IDIs) were conducted with maternity care providers (midwives, obstetric/gynaecology residents, integrated emergency surgical officers [IESOs] and obstetricians/gynaecologists) in private duty rooms and offices in the region's public hospitals. Four public hospitals were selected for the qualitative study. These four public hospitals (including two primary hospitals, one general hospital and one specialised hospital) were chosen for the qualitative study based on the caseload maternity care services provided and the order in which COVID-19 cases were initially reported in the Sidama region. Three different types of hospitals were selected: primary, general, and one specialised hospital that served as a referral centre for the Sidama region and the surrounding population in the Oromia region. This selection allowed for a nuanced understanding of the impact of the pandemic on various tiers of hospitals and their preparedness, response efficiency and the challenges they faced.

## Participant recruitment and sampling technique for qualitative data

Within each chosen hospital, we explained the study's purpose to the hospital medical director, chief executive director, and maternity care head, seeking their permission to conduct the research. Subsequently, two research assistants (both with an MSc in clinical midwifery) explained the study's purpose in detail to maternity care providers who volunteered to be interviewed. We used purposive sampling to recruit staff who provided maternity care both before and during the pandemic. All participants provided written informed consent prior to being interviewed. We aimed to recruit approximately 20 participants (10 midwives and 10 obstetricians/gynaecologists). Data reached saturation at 24 interviews. We conducted another four interviews to confirm that data were saturated before ending qualitative data collection.

## Collection tools and procedure for qualitative data

We developed a semi-structured interview guide comprising open-ended questions concerning the following factors: availability of and access to maternal and perinatal care; availability of adopted maternal and perinatal care guidelines related to COVID-19; availability of medical supplies and skilled healthcare personnel; and how challenges were overcome. The interview guide was piloted with midwives (N = 4) not included in this study. Two research assistants were recruited to facilitate the IDIs. The first author prepared the interview guide in English

and translated it into Amharic, the official language of Ethiopia. In-depth face-to-face interviews were carried out in Amharic by the first author. Each interview was conducted in the maternity care duty room or office when participants were not on duty, and we fully adhered to the Ethiopian government's COVID-19 prevention policy. The 28 interviews were conducted between 14 February and 10 May 2022, and each was digitally audio-recorded. Interview duration was approximately 30 minutes.

## Qualitative data processing and analysis

The audio recordings were transcribed immediately and listened to iteratively. Simultaneously, bilingual researchers transcribed and translated transcripts into English to check consistency. The transcriptions were imported into NVivo software (QSR International, version 12 Plus) to manage the overall data analysis. Thematic analysis [39] was employed to identify, analyse, and report themes and subthemes. We used inductive thematic analysis and followed six phases: phase 1—data familiarisation and writing familiarisation notes; phase 2—systematic coding; phase 3—generating initial themes from coded data; phase four—developing and reviewing themes; phase 5—refining, defining, and naming themes; and phase 6—writing the report [39]. All authors reviewed the themes and subthemes in the thematic analysis phases (from coding to writing a report) and approved the final themes. This study is reported according to the Standards for Reporting Qualitative Research [40] (*S1 Table*) to ensure that essential details are reported and the thematic analysis is of sufficient quality [41].

## Ethics approval and consent to participate

An internal research review board (IRB) at Hawassa University granted ethical clearance, and the University of Technology Sydney ethics committee approved the study with reference number IRB/029/14 (approval no, ETH22-7567) respectively. The research assistants explained the goal and advantages of the research project to each study participant during the data collection process. Before beginning the data collection process, each study participant provided written informed consent. Study participants were also informed of their full right to refuse, withdraw, or reject part or all of their roles in the study. All processes were carried out in accordance with the standards and laws outlined in the Declaration of Helsinki, and data were collected anonymously and kept confidential with the investigators.

## Quantitative results: Trends in antenatal care provision in fifteen hospitals

In the 12 months preceding the pandemic, from March 2019 to February 2020, the monthly data from public hospitals in the Sidama region showed a significant increase in uptake of ANC1 and ANC4. Specifically, the monthly estimated incidence rate ratio increased by 1% for ANC1 uptake (IRR = 1.011, 95% CI [1.007, 1.016]; $p < 0.0001$) and by 2.6% for ANC4 uptake (IRR = 1.026, 95% CI [1.019, 1.033]; $p < 0.0001$) (Table 1 and *Fig 2*). In the first three months of the COVID-19 pandemic, when it was at its peak, the monthly estimated incidence of ANC1 uptake decreased by 14% (IRR 0.863, 95%CI 0.812 to 0.918; P<0.0001), and ANC4 uptake decreased by 14% (IRR 0.858, 95%CI 0.782 to 0.942; P<0.001) (Table 1 and *Fig 2*). Overall trends during the initial six months of the COVID-19 pandemic revealed that ANC1 uptake significantly decreased by 0.7% (IRR 0.993, 95%CI 0.990 to 0.997; P<0.001) (N = 15,150), and ANC4 uptake significantly decreased by 1.8% (IRR 0.982, 95%CI 0.976 to 0.987; P<0.0001) (N = 5850) (Table 1 and *Fig 2*).

**Table 1. Trends of ANC provision and availability of essential medications before and during COVID-19 at public hospitals in the Sidama region, March 2019—August 2020.**

| Trends of ANC provision and availability of essential medication in the hospitals | Monthly incidence rate before COVID-19 as an estimated IRR 95% CI | P value | Monthly incidence rate immediately during the COVID-19 as an estimated IRR 95% CI | P value | Monthly incidence rate before COVID-19 compared with during COVID-19 as an estimated IRR 95% CI. | P value |
|---|---|---|---|---|---|---|
| ANC1 | 1.011(1.007–1.016) *** | 0.0001 | 0.863 (0.812–0 .918) *** | 0.0001 | 0.993 (0.990–0.997) *** | 0.001 |
| ANC4 | 1.026 (1.019–1.033) *** | 0.0001 | 0.86 (0.78–0.94) *** | 0.001 | 0.982 (0.976–0 .987) *** | 0.0001 |
| Overall availability of essential medication in the hospitals | 1.00 (0.987–0.1.013) | 0.970 | 1.001 (0.837–1.98) | 0.985 | 0.987(0.0.948–1.029) | 0.565 |
| Iron-folic acid tablet (Fefol) | 1.207 (0.654–2.229) | 0.547 | 1.006 (0.958–1.056) | 0.814 | 1.003 (0.875– 1.148) | 0.971 |
| Magnesium sulfate | 0.96 (0.54– 1.73) | 0.899 | 0.99 (0.95–1.038) | 0.782 | 0.99 (0.869– 1.136) | 0.925 |
| Oxytocin | 1.084 (0.611–1.922) | 0.782 | 1.002 (0.960–1.047) | 0.904 | 0.978 (0.858–1.15) | 0.735 |
| Ceftriaxone | 1.208 (0.664–2.197) | 0.536 | 1.017 (0.972–1.064) | 0.461 | 0.969 (0.846–1.109) | 0.645 |
| Normal saline | 1.248 (0.970–2.225) | 0.453 | 1.013 (0.969–1.060) | 0.561 | 0.977 (0.859–1.13) | 0.729 |

Note: * = Significant 0.05

** = Significant at 0.01

*** = Significant at 0.001, IRR = Incidence rate ratio

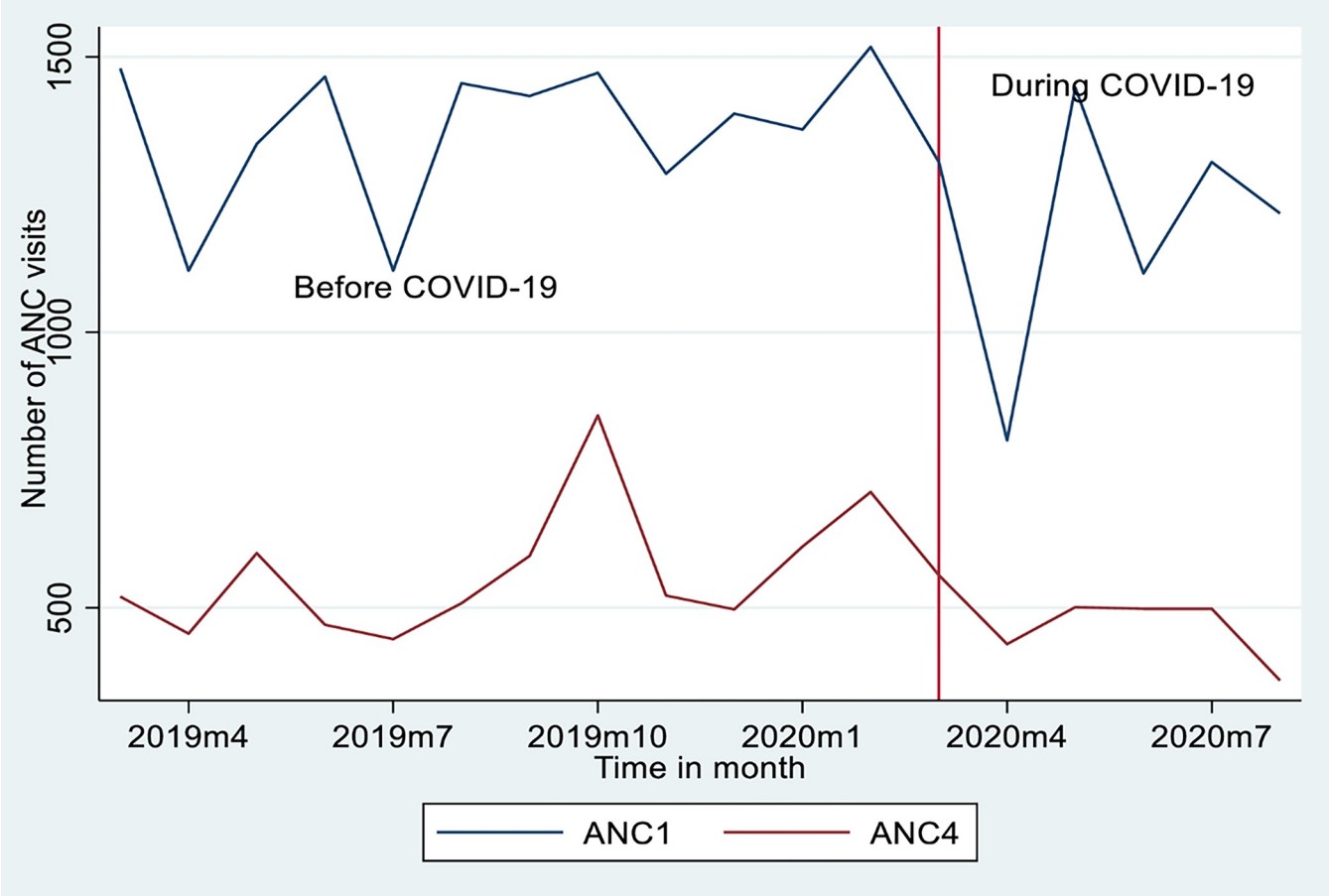

**Fig 2. Mean trends of ANC provision at the Sidama region's public hospitals in Ethiopia.**

## Qualitative results: Clinicians views and experiences of the impact of COVID-19 on antenatal care provision

In total, 28 maternity care providers were interviewed face to face: 15 midwives, 2 IESOs, 4 obstetric/gynaecology residents, and 7 obstetricians/gynaecologists (*S2 Table*). Three themes were constructed from analysis of the data, *'Barriers to ANC access during the COVID-19', 'Barriers to ANC uptake during COVID-19',* and *'Barriers to ANC provision during the COVID-19'* *Fig 3*.

Three themes were identified, namely, *'Barriers to ANC access during COVID-19', 'Barriers to ANC uptake during COVID-19',* and *'Barriers to ANC provision during COVID-19',* as displayed in *Fig 3*. Within these themes, eight subthemes were identified. In the theme of *'Barriers to ANC access during COVID-19,'* two subthemes were identified *'Shortage of resources',* and *'Community discrimination against those attending the hospital'.* Meanwhile in the theme *'Barriers to ANC uptake during COVID-19',* two subthemes included *'Fear of contracting COVID-19',* and *'Decreased attendance of ANC'.* Lastly four subthemes were identified in the theme, *'Barriers to ANC provision during COVID-19',* including *'Absence of ANC guidelines for care provision', 'Inadequate personal protective equipment (PPE)', 'COVID-19 vaccine hesitancy'* and *'Long waiting times for ANC'.* The explanations of each theme and subtheme are supported by direct quotes from study participants.

### Barriers to ANC access during COVID-19

Two subthemes emerged from our analysis of interview responses relevant to ANC access during COVID-19: *'Shortage of resources'* and *'Community discrimination against those attending the hospital'.*

### Shortage of resources

Participants indicated that a lack of resources during COVID-19 affected women's access to ANC. In particular, women's incomes declined, and public transport became more limited and doubled in cost. There was a shift of hospital resources towards COVID-19 prevention and treatment, but a lack of medical supplies, as evidenced by the following quote:

> '*The taxi cost increased during COVID-19, and the women who live in rural areas did not come to the hospital due to transport costs. For example, a woman who came from another city paid double, and she might not come to the hospital [again]'* (Midwife RMP6).

Maternity care providers described how the price of medical supplies substantially increased due to shortages, to the extent that it was impossible for most people to buy them. The combination of Ethiopia's socio-political situation and the COVID-19 pandemic prevented foreign aid from funding public hospitals, as illustrated by the following quote:

> '*Medical supplies were tough [expensive]; for example, one glove was sold for up to 100 Ethiopian birr [$2 USD]. Foreign aid has decreased due to COVID-19 and the ongoing war in the country. The NGOs donated a lot of supplies [for women's use] before COVID-19. NGOs ceased contributing medical supplies. . . no medical supplies in the hospital. Due to the economic crisis, women cannot afford to pay [$2USD] for a single glove'* (IESO ALTP14).

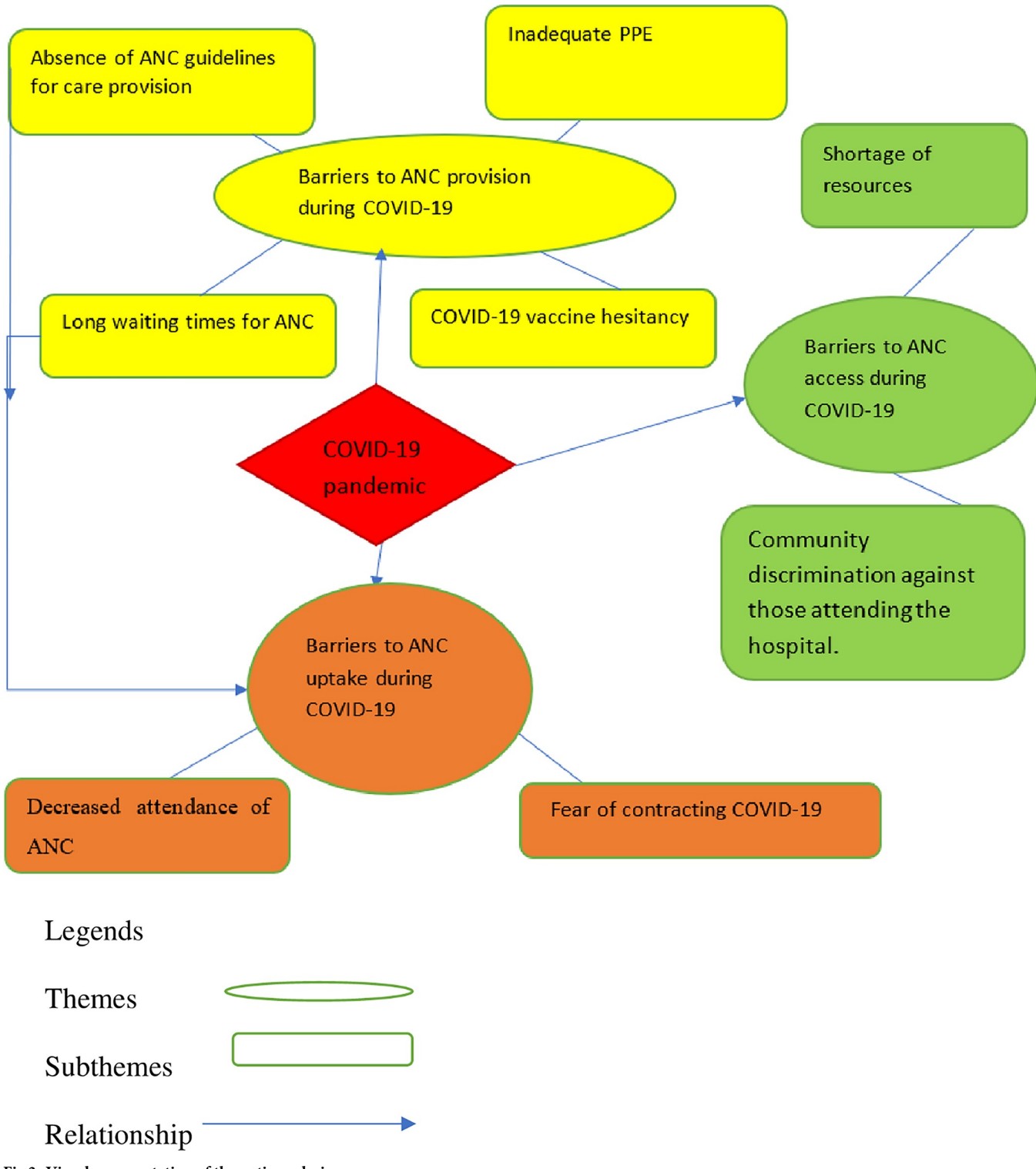

**Fig 3. Visual representation of thematic analysis.**

## Community discrimination against those attending the hospital

Discrimination by the community against individuals attending the hospital had a detrimental impact on women's ability to access ANC during COVID-19. Midwives stated that the community discriminated against those who attended hospitals, including both providers and receivers of care. Many community members believed those who visited the hospital could bring the virus to the community. This discrimination also affected healthcare providers, as they could not meet with their families or maintain their daily routines. The discrimination against maternity care providers is illustrated in the following quote:

'*We [health care providers] could not find food to eat or meet basic needs because of discrimination us in the town. Women and families discriminated against healthcare providers; they did not want us to live with them as we were involved in providing care in the hospital; they believed we could bring COVID-19 to the community, and they think COVID-19 is a killer*' (Midwife MALTP17).

'*It was a tough time; in my residential area, nurses and physicians could not obtain basic necessities and could not rent a place as a result of discrimination*' (Midwife WR1).

Discrimination resulted in a decline in social capital between healthcare providers and the community, resulting in limited access to ANC As a consequence, COVID-19 further damaged cultural cohesiveness between the community and healthcare providers, as exemplified by the following quote:

'*Social cohesion between healthcare providers and the community has broken down during COVID-19; for example, we [healthcare providers] could not attend funeral ceremonies. We are confined in the home. We live in a small town; everyone knows the healthcare providers, and they discriminated against us*' (Midwife MLP5).

Midwives reported similar discrimination against women who received care in hospitals. The community discriminated against women who sought any care at the hospital because they believed that anyone visiting the hospital would contract the virus and spread it around. Consequently, this discrimination resulted in limited access to ANC. As one midwife highlighted,

'*Pregnant women's rapport with neighbours was affected when they came to [access] care in the hospital during COVID-19. After a [pregnant] woman returned home, her neighbours discriminated against her since she was [accessing] care in the hospital, and her [neighbours] believed she had contracted COVID-19*' (Midwife MAP14).

## Barriers to ANC uptake during COVID-19

Two subthemes were identified within '*Barriers to ANC uptake during COVID-19*' theme: '*Fear of contracting COVID-19*' and '*Decreased attendance of ANC*'.

### Fear of contracting COVID-19

The midwives highlighted that a significant number of women did not attend hospitals because they feared contracting COVID-19. Due to this fear, even women who attended and received care kept their distance from maternity care providers, believing that '*the hospital was the epicentre of the virus*' (Midwife MRP1). Other midwives concurred:

'*Women came to [uptake] care in the hospital with fear. They feared healthcare providers. They considered healthcare providers to be a source of COVID-19, ha ha ha*' (Midwife MALTP16).

'*The women were afraid; as I told you before, they believed that if they came to the hospital for ANC, they would catch COVID-19*' (Midwife MLP5).

Midwives explained that pregnant women's fear of contracting COVID-19 considerably affected practitioner–client communication during care provision. Maternity care providers felt that this fear prevented them from developing a strong rapport with women or spending sufficient time in consultations. Physical-distance policies stipulated that individuals must remain at least 1.5 metres apart. In addition, because of the perceived risk to their privacy resulting from physical distancing and weakened communication, women were reluctant to disclose their medical information to maternity care providers when attending the hospital. One midwife has the following explanation:

'*During the pandemic, fear reduced women's communication with midwives. Women needed to discuss their private [sexual] issues by approaching us [healthcare providers], but distance made it impossible. When women came, we comforted them by touching their shoulders, but now it is not easy. It reduces something that we have had. Furthermore, we would keep our distance from them*' (Midwife MAP12).

Obstetricians and obstetric/gynaecology residents also described how the pandemic disrupted their relationships and communication with women. For fear of spreading the infection, at the peak of COVID-19, none would even touch a woman's medical chart, and the fetal heartbeat was checked using a Doppler rather than a Pinard fetoscope. One resident commented:

'*The rapport between women and physicians clearly declined [during COVID-19]. You could not even touch a medical chart when COVID-19 was at its peak because of the news from Italy that physicians had contracted and died from COVID-19. The disease is transmitted by touching a medical chart. There was a great distance. As a result, there was a decline in the rapport between women and physicians. If physicians were doing these things, you might assume other healthcare providers were as well, and many things were missed in giving care*' (Resident 4 RRP10).

'*Maternity care providers' rapport with women substantially decreased. For example, before the pandemic, we used* Pinard fetoscope *to check the fetal heartbeat. Nevertheless, there were numerous concerns about using a* Pinard fetoscope *to check a fetal heartbeat during the pandemic, so we switched to a Doppler to check the heartbeat. The maternity care providers believed that women with a cough had COVID-19, and so were unwilling to care for them*' (Obstetrician SRP19).

### Decreased attendance of ANC

Reporting a decrease in ANC attendance during COVID-19, participants noted that women perceived the hospital as not providing the services and failing to address their needs. In addition, maternity care providers were not actively encouraging women to visit the hospital, leading to recommendations for staying at home, as noted in the following quote:

'*After Ethiopia reported the first COVID-19 case, maternal health services significantly decreased, and the number of women who came to the hospital, especially for their ANC follow-up, decreased because they were listening to other news about COVID-19 through various media outlets. Many women were staying at home. They believed that the hospital was not providing the services and did not take their needs into consideration when they came*' (IESO ALTP15).

Due to population growth, ANC attendance is expected to grow from one year to the next. However, during COVID-19, obstetricians reported declining daily ANC attendance: '*women did not come on their appointment day*' (Obstetrician SAP22). One obstetrician expressed an opinion that the decline in ANC attendance was in part due to the messages recommending the community stay at home, and the fact that women were not encouraged to access the services by medical and midwifery staff at the hospital. Further evidence is provided by the following quote:

'*Before COVID-19, there were around ten women [per day], but during COVID-19, it was around five to seven women. Our population is increasing, but the number of women attending decreased during COVID-19. They came, we treated them, sent them home, and made predictions. As I had predicted, fewer people attended hospitals. According to my predictions, these decreases resulted from the fact that we did not encourage women to attend the hospital . . .there was a lack of maternity care providers in this hospital*' (Obstetrician SRP24).

## Barriers to ANC provision during COVID-19

Four subthemes were identified within the theme '*Barriers to ANC provision during COVID-19*': '*Absence of ANC guidelines for care provision*', '*Inadequate PPE*', '*COVID-19 vaccine hesitancy*', and '*Long waiting times for ANC*'.

## Absence of ANC guidelines for care provision

Participants acknowledged that the availability of ANC guidelines is crucial for providing high-quality care and evidence-based care. Most study participants reported that there were no COVID-19-related ANC guidelines in the hospitals. One obstetrician said, '*We did not have any protocol. . . We continued to use the previous obstetric care guidelines*' (Obstetrician SRP 23). While there was some guidance from the WHO on emergency infection prevention, there was no specific protocol for the management of pregnancy care:

'*There is no unique protocol or approach that we adapted or adopted. The hospital got some aid from the regional health bureau and gave us essential personal protection equipment like gloves, masks, and sanitiser. Using this equipment, we admitted the women who came to the service. However, we did not prepare any COVID-19 protocol or design any strategies at the curriculum level*' (IESO ALTP15).

'*The WHO developed the guideline and then nationally adopted it. By chance, we have one guideline that was adopted and updated from the previous national guideline in obstetrics in 2020. However, the guideline was not related to COVID-19. Meanwhile, there is a protocol for emergency infection prevention and COVID-19 prevention protocol. It is used for all, including pregnant women. However, specifically, there is no guideline for pregnancy-related care during COVID-19*' (Resident 4 RRP 10).

### Inadequate PPE

Midwives indicated that inadequate supplies of COVID-19 PPE were a common problem in hospitals during the pandemic. Maternity care providers noted an imbalance between demand and supply of PPE in the hospitals. The scarcity of *'facemasks, face shields, shoes and other PPE'* (Midwife MAP 13), perhaps due to increased use of infection control supplies. The following quotes are illustrative:

> '*Five masks were given to each healthcare provider per week, and one sanitiser was given to many healthcare providers, but it was not given individually. [Later] the number of masks was reduced from five to four per week'* (Midwife MALTP17).

> '*Women bought a mask and used it together; when someone left the hospital, they handed it over to another person'* (Midwife MLP5).

### COVID-19 vaccine hesitancy

At the time of data collection, only 9 of the 28 participating maternity care providers had received two doses of the COVID-19 vaccination. If maternity care providers are hesitant about the COVID-19 vaccine, it may impact pregnant women's confidence in the vaccine's safety and efficacy. This could lead to lower vaccine rates and hinder ANC provision. Some mentioned that they doubted the vaccine's efficacy or believed that the infection was a punishment for disobeying God's commandments, as exemplified in the following quote:

> '*I have chosen not to get vaccinated. I do not believe in it, and I do not want the vaccine. My belief is that the outbreak of COVID-19 may be a consequence of our sins was caused by our sins and evil actions. I consider COVID-19 to be a form of divine punishment for our transgressions. Therefore, I believe we should fast and pray in an attempt to avoid this disease'* (Midwife MAP13).

### Long waiting times for ANC

Waiting times for ANC provision were longer than usual during COVID-19. Priority was given to women wearing facemasks; hence, those without facemasks had to wait longer. As one midwife commented, *'The women who wear facemasks get [to access] the service first; those who do not wear facemasks do not get [access to] care'* (Midwife MALTP17). Moreover, maternity care providers were unable to provide the service without face masks, causing delays in providing ANC in hospitals when facemasks were unavailable in the ANC clinic. Women who did not wear facemasks were not permitted to enter ANC clinics or receive immediate care. Despite the decline in ANC attendance, those women who did attend were met with delays in receiving care due to lack of staff, as illustrated by the following quote:

> '*We were nagging patients who did not wear masks, we ordered a woman to wear a mask, and we left the room if she did not wear a facemask. However, we offered ANC during COVID-19 in a manner similar to that before the pandemic; all ANC components were offered during [COVID-19]. Although we performed complete physical examinations. . .comparable to those performed before COVID-19, the facemask did not bring comfort to women or those who provide maternity care and had a terrible effect during physical examinations'* (Midwife MALTP16).

*'Many women were waiting [to access] the services, crowded on ANC waiting chairs, and the waiting time was long because [only] one or two maternity care providers gave services. Women were desperate to get [access to] the service, and they [sometimes] returned to their homes without getting healthcare. We [used to provide] ultrasounds for all women during each visit. During COVID-19, we [only] performed ultrasounds for selected women'* (Obstetrician SRP23).

## Integrating quantitative and qualitative findings using joint display technique

In the quantitative analysis, monthly trends of ANC attendance and the availability of essential drugs in hospitals were assessed before and during the pandemic. Simultaneously, three themes and eight subthemes were also identified in the qualitative data to explore whether the qualitative findings confirmed or disconfirmed the quantitative findings.

For ANC1 and ANC4 attendance, the quantitative findings revealed decreasing trends during the pandemic, and the qualitative study corroborated these declines, providing evidence that stay-at-home recommendations, lack of (and more expensive) transport, and fear of contracting the virus contributed to reducing the number of women attending hospitals for ANC (Table 2). As one participant commented, *'the uptake of ANC4 declined due to fear of contracting the virus during COVID-19'* (Midwife MLP5).

By contrast, the quantitative and qualitative findings on essential drug availability were contradictory (Table 2). The quantitative analysis showed no significant difference in the availability of essential drugs in hospitals before and during the pandemic. However, access was impacted for women by the cost, and according to staff, by supply shortages, as the following quote demonstrates, *'medical supplies decreased in the hospital, and the cost of medical supplies increased. All things were going up during the COVID-19 pandemic'* (IESO ALTP14). The same discrepancy was found for iron and folic acid availability, which did not significantly differ between before and during the pandemic according to the quantitative analysis but notably declined according to the qualitative analysis: *'pregnant women faced challenges in accessing and taking iron-folic acid supplement appropriately during the pandemic' (*Midwife MALTP18). These inconsistencies could be explained by inadequacies in the HMIS inventory. The binary nature of data in the HMIS form means that responses are limited to 'available' or 'unavailable', with no space to include details of the quantities of essential drugs. Hence, the HMIS data do not adequately represent the scenario on the ground at the hospital pharmacy level (Table 2).

There is a lack of data to demonstrate quantitatively whether ANC guidelines are in place, the amount of PPE equipment available or vaccine rates to confirm qualitative findings. Additionally, there is no data to determine wait times for ANC, presence and severity of fear of contracting COVID-19, or measures of discrimination from the community. This information reported by care providers is similarly reported in the literature for other comparable countries (Table 2).

## Discussion

Understanding the difficulties of access to ANC during any disease outbreak is essential to ensuring and sustaining ANC services [42]. This understanding aids in the identification of specific barriers within both the community and the healthcare system during a crisis, providing opportunities to mitigate their indirect consequences. This study demonstrated that access

**Table 2. Joint display of quantitative and qualitative findings for each theme and subtheme on impact of COVID-19 on ANC access, uptake and provision, and mixed-methods meta-inferences.**

| Theme | Subtheme | Quantitative findings | Qualitative findings | Mixed-methods meta-inference |
|---|---|---|---|---|
| Barriers to ANC access during COVID-19 | Shortage of resources | Overall trends of availability of essential drugs before and during COVID-19 showed no significant change (IRR = 0.987, 95% CI [0.948, 1.029]; $p = 0.565$) | Maternity care providers demonstrated a shortage of resources, and the women could not afford to buy medical supplies in the hospital while consistently lacking essential medical supplies. The quote below illustrates: *'There was a shortage of medical supplies during COVID-19; for example, in ANC clinics, iron folic acid and TT vaccine were unavailable'* (IESO ALTP 15). | Contradiction: the discrepancy between quantitative and qualitative findings could be explained by the binary responses (i.e., available or unavailable) in the HMIS form, providing no insight into the exact quantities of essential drugs. By contrast, the qualitative findings explored maternity care providers' day-to-day experiences of access to essential medical supplies in the hospital. The shortage of essential drugs, such as iron and folic acid, may have caused difficulty for women in accessing ANC. |
| | Community discrimination against those attending the hospital | None | Maternity care providers reported that women who sought care at a hospital during the pandemic often faced discrimination from the community, which hampered women access to ANC. The quote below illustrates: *'There is discrimination. If women went to the hospital, the community assumed they would come back with COVID-19. Women may have been psychologically let down because there was no longer the same social cohesion as before COVID-19* (Midwife MAP14). | The qualitative findings demonstrate that discrimination contributed to reducing ANC access. With a decline in social cohesion, pregnant women feared facing community discrimination after visiting hospital, thus affecting ANC access during COVID-19. |
| Barriers to ANC uptake during COVID-19 | Fear of contracting COVID-19 | None | Fear of contracting COVID-19 was mentioned as a barrier to the uptake of ANC by maternity care providers. The following quote observes: *'The outpatient department was not as crowded during COVID-19 as it was before COVID-19 because people stayed home due to fear of contracting the virus'* (Midwife WR1). | The qualitative findings indicated that fear of contracting COVID-19 contributed to reducing ANC uptake. Many pregnant women were more afraid of contracting COVID-19 disease than of suffering pregnancy-related complications, leading to a decline in ANC uptake during the pandemic. |
| | Decreased attendance of ANC | Overall trends before and during COVID-19 showed significant reduction of 0.7% in ANC1 provision (IRR = 0.993, 95% CI [0.990, 0.997]; $p < 0.001$), and significant decrease of 1.8% in ANC4 provision (IRR = 0.982, 95% CI [0.976, 0.987]; $p < 0.0001$) | Maternity care providers described a decline in ANC attendance during COVID-19. *'Pregnant women did not come to their appointments for ANC follow-up. We tried to reinstate ANC follow-up before COVID-19, but they stayed at home; we tried to provide care to them via phone, but it was challenging'* (Obstetrician SAP22). | Corroboration: the qualitative and quantitative findings indicate that recommendations to stay at home led to pregnant women postponing or cancelling ANC appointments, this reducing ANC uptake during the pandemic. |

*(Continued)*

**Table 2.** (Continued)

| Theme | Subtheme | Quantitative findings | Qualitative findings | Mixed-methods meta-inference |
|---|---|---|---|---|
| Barriers to ANC provision during COVID-19 | Absence of ANC guidelines for care provision | None | Maternity care providers reported that no guidelines were introduced for maternal care during the pandemic, notwithstanding ANC appointment changes made by the health bureau. '*There were no maternal care guidelines related to COVID-19. Meanwhile, appointment intervals were changed by the health bureau*' (Midwife RMP221). | The qualitative findings indicate that the absence of ANC guidelines during COVID-19 reduced the provision of ANC that leading to quality-of-care provision was suboptimal. |
| | Inadequate PPE | None | Maternity care providers reported that women and healthcare providers used inadequate COVID-19 PPE. '*There was a shortage of materials used to prevent COVID-19, for example, soap, hand sanitiser, and masks*' (Midwife MLP5). | The qualitative findings illustrate that inadequate PPE was a barrier to ANC provision. Maternity care providers could not provide optimal ANC without adequate PPE. |
| | COVID-19 vaccine hesitancy | None | Some maternity care providers mentioned doubts over the efficacy of COVID-19 vaccines, and beliefs that the vaccine causes various diseases, including clotting disorders. '*It causes blood clotting, cancer, and other issues. There are no diseases more dangerous and fatal than cancer than blood clotting and cancer*' (Midwife MAP14). | The qualitative findings demonstrate that COVID-19 vaccine hesitancy impacted ANC provision. Such hesitancy in maternity care providers could have increased the reluctance of pregnant women to be vaccinated, since the providers would not attempt to persuade them of the vaccine's benefits during pregnancy. Furthermore, vaccine-hesitant maternity care providers' fear of contracting the virus when providing ANC could have further reduced ANC provision. |
| | Long waiting times for ANC | None | Maternity care providers commented that women faced long waiting times for ANC during the pandemic, especially if they did not wear a mask. '*The women who wear facemasks get [to access] the service first; those who do not wear facemasks do not get [to access] care until they bought and wore a mask*' (Midwife MALTP17). | The qualitative findings illustrate that long waiting times hindered ANC provision during COVID-19. Maternity care providers could not allow care to pregnant women until they wore a facemask, leading to reduced ANC provision during the pandemic. |

to ANC was disrupted due to a shortage of resources and discrimination against those attending the hospital during COVID-19. These findings are consistent with a systematic review conducted in three West African countries during the Ebola outbreak [43], showing that disruption to ANC access during the Ebola outbreak was the result of community mistrust of the health facilities and discrimination against those who attended and provided ANC in the health facilities. Such disruptions could lead to a weakened coherence between the community and healthcare providers. A qualitative study in rural India found that ANC access was limited in the first wave of COVID-19 by lockdown restrictions and a prevailing sense of mistrust in the public health system and its functioning [42]. A studies in Kenya, Nigeria and India [15, 21, 24, 25] align with this findings that ANC access was restricted by lack of transport or inability to afford it, inability to pay for medical expenses, and the closure of non-essential services during the pandemic.

Quantitative findings showed that the availability of essential medical supplies before and during COVID-19 remained stable. The qualitative study, however, revealed that most study participants observed a shortage of essential medical supplies. Before COVID-19, maternity care providers reported that all pregnant women received care without any charge and were

supplied with essential supplements. For over a decade, a pillar of the Ethiopian government's programme to reduce maternal and neonatal mortality rates was the provision of free care to pregnant women and newborns in public health facilities. This had made a significant impact in lowering maternal and neonatal mortality, but the additional burden of COVID-19 is a possible threat to the free care that supported this progress. The burden on the health system resulting from the pandemic might raise the costs and demand for medical supplies, for example, increased usage of PPE, gloves, essential drugs, and increased prices, triggering an economic downturn that could continue to affect ANC access [44, 45].

The challenges of accessing ANC during the pandemic resulted in decreased uptake, consequently reducing the proportion of women obtaining at least a minimal level of evidence-based, routine ANC and management of complications during ANC at healthcare facilities. In 2016, the WHO recommended that pregnant women have at least eight ANC contacts with maternity care providers throughout their pregnancy [46]. Prior to the pandemic, Ethiopia had only recently implemented this guideline and was struggling to increase ANC4 coverage. The unprecedented impact of COVID-19 has placed a double burden on the country's maternal healthcare system, directly and indirectly affecting ANC uptake.

This study found that during the first six months of the pandemic, mean monthly incidence rates of ANC1 and ANC4 uptake declined considerably. These quantitative findings were corroborated by the qualitative findings. According to maternity care providers, women's attendance for ANC follow-up fell at the COVID-19 peak in April 2020. Maternity care providers demonstrated that women's fears of contracting either virus were a barrier to ANC uptake. These findings also align with a literature review from three West African countries [47], and a qualitative study exploring how the fear of contracting Ebola [48] hindered the uptake of ANC.

Our findings demonstrate that the lack of ANC guidelines, long waiting times, inadequate PPE, and COVID-19 vaccine hesitancy affected ANC provision during the pandemic. These findings coincide with the results of a study in Nigeria [24], which showed that long waiting times, shortages of medical supplies and human resources, and healthcare providers' lack of preparedness were barriers to ANC provision during the first wave of COVID-19.

In this study, maternity care providers explained that during the pandemic, there were no guidelines adapted to the provision of ANC and intrapartum care in hospitals during COVID-19. However, revised guidelines for the national comprehensive COVID-19 clinical management handbook [49] were issued in 2020. This updated protocol placed more emphasis on the management of COVID-19-suspected and confirmed pregnant women during ANC and intrapartum care.

In addition, the existing obstetric management protocol guidelines for hospitals were revised and issued in 2021 [50]. While these updated hospital protocols address ANC and intrapartum care, they do not make explicit recommendations for service delivery during a pandemic. This finding is inconsistent with a study in LMICs that showed how ANC guidelines were adopted and locally tailored for familiarisation by maternity care providers to preserve and improve the provision of ANC during COVID-19 [18]. However, another global cross-sectional study of 714 maternal and neonatal professionals, 39% of whom were in LMICs, found that 53% of study participants received no updated guidelines during the pandemic [51].

In this study, participants indicated that a shortage of PPE supplies during the pandemic made it difficult to provide ANC. These findings corroborate the results of a study in Nigeria, which found that a lack of PPE inhibited maternity care provision [24]. A lack of COVID-19 prevention materials in hospitals could have reduced the rapport between pregnant women and maternity care providers. According to participants, there were frequent shortages of non-pharmaceutical prevention supplies, including facemasks, PPE, gloves, face shields, and sanitisers. This finding coincides with a study in Nepal [52] found that lack of PPE affected ANC

provision.Our results also align with the findings of a global survey exploring how wearing facemasks affected gesture communication and facial expressions between patients and health-care providers [51].

Vaccine hesitancy among healthcare providers is a factor in preventing greater coverage of population immunity and affecting the provision of ANC. Mistrust of the COVID-19 vaccine was expressed by maternity care providers in this study. A cross-sectional study conducted in Nigeria [53] and Egypt [54] showed that respectively, 50.5% and 42% of healthcare workers exhibited vaccine hesitancy. Vaccine hesitancy may have been related to healthcare providers' fears of adverse effects in their future pregnancies and other medical complications, efficacy uncertainty, inadequate vaccine trials prior to human administration, doubts about vaccine benefits, misinformation and disinformation about the vaccine's side effects [55].

Participating maternity care providers reported the limited availability of iron and folic acid during the pandemic. This is consistent with the finding of a study conducted in Northwest Ethiopia exploring the unavailability in hospitals of ferrous sulphate 150 mg + folic acid 0.5 mg tablets, which are routinely prescribed to pregnant women for at least three months in areas where iron deficiency anaemia is common [56].

Our findings showed that the shocking declines in ANC uptake and provision reinstated after four months. This could be attributed to the actions of the Minister of Health, who implemented various strategies to ensure the continuation and maintenance of essential services, including raising the number of healthcare providers at health facilities and establishing a non-COVID-19 task force to reverse the decline in ANC attendance [57]. Hospitals also carried out home visits and made phone calls to pregnant women with registered phone numbers in order to boost ANC uptake among pregnant women.

The strength of our study lies in our mixed-methods approach and collection of data from 15 public hospitals across the Sidama region. We explored the perceptions of a diverse range of maternity care providers through IDIs.

However, the study also has limitations. Quantitative data were obtained from the HMIS, an administrative data source that could contain inaccuracies: it is possible that relevant data were overestimated or underreported, especially in the absence of population-level denominators. Staffing issues during the pandemic might have impacted the accuracy of HMIS reporting. Since our study primarily examines hospital-level data, further research is needed to assess ANC access, uptake, and provision at primary health centres.

## Conclusion and recommendations

Our findings demonstrate that the COVID-19 pandemic affected ANC access, uptake, and provision in the study area from March to August 2020. To mitigate disrupted ANC access, uptake and provision, ANC clinics should be equipped with medical supplies. It is crucial to maintain rapport between the community and maternity care providers and provide training for maternity care providers regarding the adapted/adopted guidelines during COVID-19 at the hospital grassroots level for use in the current and future pandemics. Pregnant women should have access to timely care from maternity care providers, to maintain at least a minimum standard of care in the current and future pandemics. Further studies are needed to understand the long-term impact of COVID-19 on ANC access, uptake, and provision, as well as the impact on these factors of internal conflict within Ethiopia.

## Supporting information

**S1 Table. Standards for Reporting Qualitative Research (SRQR).**
(DOCX)

**S2 Table. Sociodemographic characteristics of study participants (N = 28).**
(DOCX)

## Acknowledgments

First, we would also like to thank study participants, health bureau officials, HMIS data managers and hospital chief executive directors for cooperating during the study. The author would like to acknowledge Dr Caroline Havery for her help with English grammar.

## Author Contributions

**Conceptualization:** Zemenu Yohannes Kassa, Vanessa Scarf, Sabera Turkmani, Deborah Fox.

**Data curation:** Zemenu Yohannes Kassa, Vanessa Scarf, Sabera Turkmani, Deborah Fox.

**Formal analysis:** Zemenu Yohannes Kassa, Vanessa Scarf, Sabera Turkmani, Deborah Fox.

**Funding acquisition:** Zemenu Yohannes Kassa, Vanessa Scarf, Sabera Turkmani, Deborah Fox.

**Investigation:** Zemenu Yohannes Kassa, Vanessa Scarf, Sabera Turkmani, Deborah Fox.

**Methodology:** Zemenu Yohannes Kassa, Vanessa Scarf, Sabera Turkmani, Deborah Fox.

**Project administration:** Zemenu Yohannes Kassa, Vanessa Scarf, Sabera Turkmani, Deborah Fox.

**Resources:** Zemenu Yohannes Kassa, Vanessa Scarf, Sabera Turkmani, Deborah Fox.

**Software:** Zemenu Yohannes Kassa, Vanessa Scarf, Sabera Turkmani, Deborah Fox.

**Supervision:** Zemenu Yohannes Kassa, Vanessa Scarf, Sabera Turkmani, Deborah Fox.

**Validation:** Zemenu Yohannes Kassa, Vanessa Scarf, Sabera Turkmani, Deborah Fox.

**Visualization:** Zemenu Yohannes Kassa, Vanessa Scarf, Sabera Turkmani, Deborah Fox.

**Writing – original draft:** Zemenu Yohannes Kassa, Vanessa Scarf, Sabera Turkmani, Deborah Fox.

**Writing – review & editing:** Zemenu Yohannes Kassa, Vanessa Scarf, Sabera Turkmani, Deborah Fox.

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
