## [Decision Letter · Decision Letter 0]

1 Jun 2023

PONE-D-23-07658Impact of COVID-19 on antenatal care provision at public hospitals in Ethiopia: a mixed method studyPLOS ONE

Dear Dr. Kassa,

Thank you for submitting your manuscript to PLOS ONE. After careful consideration, we feel that it has merit but does not fully meet PLOS ONE’s publication criteria as it currently stands. Therefore, we invite you to submit a revised version of the manuscript that addresses the points raised during the review process. Major revision of your manuscriptis require before further consideration. Psy atten to reviewer 2 comments and make appropriate corrections.

We look forward to receiving your revised manuscript.

Kind regards,

Ephraim Kumi Senkyire

Academic Editor

PLOS ONE

Journal Requirements:

Reviewers' comments:

Reviewer's Responses to Questions

**Comments to the Author**

1. Is the manuscript technically sound, and do the data support the conclusions?

Reviewer #1: Yes

Reviewer #2: No

2. Has the statistical analysis been performed appropriately and rigorously? 

Reviewer #1: Yes

Reviewer #2: No

3. Have the authors made all data underlying the findings in their manuscript fully available?

Reviewer #1: Yes

Reviewer #2: Yes

4. Is the manuscript presented in an intelligible fashion and written in standard English?

Reviewer #1: Yes

Reviewer #2: No

5. Review Comments to the Author

Reviewer #1: Thank you for the good work.

1. the first theme is described as "the impact of ANC access"........I hope it is a typing error and better to write it as" access to ANC or the impact of COVID-19 on ANC access...."

2. please remove the last row in Table 3(Joint display of quantitative, qualitative and mixed methods results ...)

Reviewer #2: Comments

Thank you for your invitation to review this paper. Here are comments listed below:

Title

The study is already over searched, Even a systematic review of metanalaysis have been done in Ethiopia entitled with <impact a="" and="" covid-19="" essential="" ethiopia:="" healthcare="" in="" maternal="" metaanalysis="" of="" on="" pandemic="" review="" services="" systematic="" utilization=""> or link https://journals.plos.org/plosone/article?id=10.1371/journal.pone.0281260#:~:text=Impact%20of%20COVID%2D19%20pandemic%20on%20ANC%20services,CI%3A%2015.85%2C%2022.76) . So what you added to your finding?

The title should be rewrite in SMART form meaning that mainly it should clear where (specifically for local reader better to add < Sidam Region, Ethiopia> and when the study done.

Abstract

Methods section

What is the exact data collection period? What is your sample size? It is not clear to the reader. The author collect data before covid 19 (March 2019 to February 2020), so how to relate the study with covid 19?

Result

No information mentioned about the quantitative data in your result. So please include them.

Conclusion

Authors should be selective to generalize the whole finding what they have gotten from the finding. It is not clear that why the authors select only three views (ANC access, uptake and provision) for conclusion?

Keywords

The author should select MESH terms for key word selection.

Introduction

The first sentence needs citation–page 2, line49-50

Please avoid the use of reference repeatedly. E.g. reference 1, 3, etc. repeated more than two times.

Page 3 line 69, insert citation of reference (9) at the end of sentence.

Methods

Please Rewrite as <methods and="" materials="">

Setting

All information presented here are mentioned in the methods section. So please try to merge it or delete one them since in scientific paper writing form no need of redundancy.

What are your study settings/institutions? It is not clearly stated, and try to focus on your study area and selected settings.

Quantitative methods

Setting

Please rewrite the whole paragraph of this section. No need of writing unnecessary information in study setting like the first sentence you wrote about study design <we an="" implemented="" interrupted="" series="" time="">.

Sample size not determined. Why?

Data processing and analysis

It lacks focus. It should describe about data entry and analysis, but not data collection methods, data collection procedures etc.

Qualitative methods

Please omit redundancy. Your method part is not clear as a whole. You have three section methods in this document (methods, quantitative methods, and Qualitative methods). You should have one comprehensive methods or a max of two sections of methods.

Study participant recruitment and sampling technique

Who is your research assistant? Please clearly mentioned them

What are your study participants? Patient or health care provider

Why the authors used a purposive sampling technique? Is that good to use non probable sampling methods to write scientific papers important for scientific evidence? In my view it is not acceptable.

Please insert this sentence in <participants being="" consent="" informed="" interviewed="" prior="" provided="" to="" written=""> in Ethical approval and consent section.

What about Study participant recruitment and sampling technique of quantitative methods??

Data collection tools and procedure

No information mentioned here that describe about the data collection instrument/ tool you used in measurable way. Where the questionnaire you adapted or adopted? Is the too you used is valid or not? If valid what is its validity and reliability test value?

Data processing and analysis

It seems only qualitative study design study. What about Data processing and analysis of quantitative section?

Results

Trends in antenatal care provision in fifteen hospitals

Where you have got IRR 1.1%? Please revise it (1%). Line 227

Qualitative results

Please rewrite the participants’ response in quotation form.

Where is the result of the quantitative part? You have mentioned only the qualitative and joint result. Please add the missed section.

Discussion

Try to discuss both the quantitative and qualitative findings.

Please revise and rewrite in two side comparative form.

Conclusion

Replaced conclusion with < conclusions and recommendations>

Try to focus on your finding

e.g. You recommended <additional are="" country="" factors="" impact="" in="" long-term="" needed="" on="" ongoing="" studies="" the="" these="" to="" understand="" war="">. It is out of your scope of study/title.

As a whole please try to contact English language expertise and revise the grammar, punctuation, syntax etc. of the entire document to be clear, easily understandable to the readers.</additional></participants></we></methods></impact>

6. PLOS authors have the option to publish the peer review history of their article (what does this mean?). If published, this will include your full peer review and any attached files.

Reviewer #1: No

Reviewer #2: No

---

## [Author Response · Author response to Decision Letter 0]

16 Jul 2023

Dear reviewers, 

Thank you for taking the time to review our manuscript and for your constructive comments and suggestions, which are vital to improving the quality of the manuscript. We addressed your comments point by point, provided clarifications, and incorporated your suggestion into the manuscript as follows:

Reviewer 1 comments Thank you for the good work.

1. the first theme is described as "the impact of ANC access"........I hope it is a typing error and better to write it as" access to ANC or the impact of COVID-19 on ANC access...."

Authors response

Thank you for your comment, the revision has been made in line 52, page 3.

The impact of COVID-19 on ANC access

2. please remove the last row in Table 3 (Joint display of quantitative, qualitative and mixed methods results ...) 

Authors response

We understand your concern; however, the last row in Table 3 shows the argument about whether qualitative findings support the quantitative findings and, if yes, how and if not, why. We modified the arguments and rigorously explained the confirmation and disconfirmation between quantitative and qualitative findings in Table 3.

Reviewer 2 comments 

1. The study is already over-researched, even a systematic review of meta-analysis has been done in Ethiopia entitled with <Impact of COVID-19 pandemic on utilisation of essential maternal healthcare services in Ethiopia: A systematic review and meta-analysis> or Link 

https://journals.plos.org/plosone/article?id=10.1371/journal.pone.0281260

So, what you added to your finding? 

Authors response 

Dear reviewer, we appreciate your concern regarding the over-researched issue related to our study. To the best of our search and knowledge, no study is exploring the impact of COVID-19 on ANC access, uptake, and provision in a rigorous method. Included studies in this systematic review and meta-analysis (the link you provided) and other studies used the impact of COVID-19 on ANC utilisation; however, they are quite different. The detailed difference between our study and other studies conducted in Ethiopia is as follows:

1. Most of the studies do not show ANC access, uptake, and provision before and during the pandemic. However, they showed ANC uptake during the pandemic without comparing it to before the pandemic, and they did not explore the impact of COVID-19 on ANC access, uptake, and provision. 

2. There are also methodological differences; the current study differs from previously published studies on the analysis model employed in the quantitative study and integration of quantitative and qualitative findings. Hence, previously published studies employed a binary logistic regression model, Paired t-test and non-parametric (Wilcoxon signed-rank) tests to estimate the trends of ANC uptake before and during the pandemic. However, the current study used interrupted time series analysis to estimate the trend of ANC uptake before and during the pandemic based on the recommendations from three biostatisticians. As described in the methods section of this paper, the rationale for using the interrupted time series analysis is found in lines 148-151 of page 6. Interrupted time series analysis is a robust statistical analysis method that recommends evaluating the impact of population-level interventions, including policy changes, infection prevention programmes, and pandemics like COVID-19 that have been experienced at a clearly defined time. 

3. As described in lines 154-155, page 6, we used the Poisson regression model because the outcome of interest is count data. Furthermore, in the current study, we controlled seasonal and other incidents' effects during the analysis. 

4. We used Braun and Clarke’s thematic analysis approach using NVivo 12 plus, which followed six phases to identify themes in our qualitative study. This has not been conducted in previously published articles from Ethiopia.

 5. Our qualitative findings explored the impact of COVID-19 on ANC access, uptake, and provision in a rigorous method, while previous studies did not rigorously explore the impact of COVID-19 on ANC access, uptake, and provision independently.

6. We integrated the quantitative and qualitative findings in a joint display analysis based on the recommendations from John W. Creswell’s book (2014) and Fetters MD (2019), which showed three types of integration for concurrent mixed methods- study data transformation, joint display technique and side-by-side (weaving) in the discussion section. In this study, we used the joint display approach, presented in Table 3; however, previous studies do not indicate which integration approach was used in concurrent mixed methods study.

7. In summary, the above justifications clearly show the difference between our current study from previously published works in the same area in Ethiopia. This study contributes to the body of evidence about critical issues and how stakeholders can maintain and enhance ANC access, uptake, and provision for future health systems in crisis related to the epidemic, pandemic, and any natural and man-made disaster. Additionally, this study adds to the body of knowledge and aims to contribute vital evidence to inform the planning and management of future population-level disasters (including epidemic/pandemic situations). 

2. The title should be rewrite in SMART form meaning that mainly it should clear where (specifically for local reader better to add < Sidam Region, Ethiopia> and when the study done. 

Authors response

Thank you for your comment, the revision has been made in line 1, page 1. 

Impact of COVID-19 on antenatal care provision at public hospitals in Sidama region, Ethiopia: a mixed method study

Abstract

3. Methods section 

What is the exact data collection period? 

Authors response 

Thank you for your comment, the revision has been made in lines 30-31, page 2. 

A concurrent mixed methods study was applied between 14 February 2022 and 10 May 2022 at fifteen public hospitals in the Sidama region.

4. What is your sample size? It is not clear to the reader.

Authors response

Thank you for your comment. Interrupted time series design focuses on any change or difference in a specific time interval due to interventions such as policy change or any disease outbreak. In our case, monthly ANC uptake significantly differs before and during the pandemic (the pandemic is considered an intervention). In our quantitative study, there is a lack of population-level denominators, and we described it as a limitation. The revision has been made in lines 31-35, page 2. 

This study included all pregnant women who uptake ANC before COVID-19 (12 months from March 2019 to February 2020) and during COVID-19 (six months from March 2020 to August 2020) at fifteen public hospitals in the Sidama region. The total number for the ANC1 cohort was 15,150, and 5,850 for the ANC4 cohort. The final dataset amounted to 21,000 women, and 28 maternity care providers were interviewed.

5. The author collect data before covid 19 (March 2019 to February 2020), so how to relate the study with covid 19?

Author response

We appreciate your concern, and our objective is to estimate the monthly attendance of women for ANC at public hospitals by comparing before and during the pandemic. This allowed us to assess monthly trends of ANC attendance and control for seasonal and other influencing effects. We extracted data from HMIS at 15 public hospitals for the monthly report before COVID-19 (March 2019 to February 2020) and during COVID-19 from March to August 2020 to estimate monthly trends of ANC uptake before and during the pandemic. As described in lines 38-41, page 2.

We used routinely collected data derived from the health management information system (HMIS) in fifteen hospitals in the Sidama region, Ethiopia. Monthly data were collected from March 2019 to February 2020 (12 months) before COVID-19 and from March to August 2020 (6 months) during COVID-19.

6.Result

No information mentioned about the quantitative data in your result. So please include them. 

Author response

Thank you for your comment, the revision has been made in lines 49-51, pages 2 & 3. 

The incidence rate of ANC1 uptake decreased by 0.7% (IRR 0.993, 95%CI 0.990 to 0 .997; P<0.0001) (N =15,150), and ANC4 uptake decreased by 1.8% (IRR 0.982, 95%CI 0.976 to 0.987; P<0.001) (N=5850) in the first six months of the pandemic.

7.Conclusion

Authors should be selective to generalize the whole finding what they have gotten from the finding. It is not clear that why the authors select only three views (ANC access, uptake and provision) for conclusion? 

Author response

We acknowledge your concern, and ANC access, uptake, and provision were identified themes in the qualitative findings and pertinent findings in this study that add new insight to the impact of COVID-19 on ANC provision in this study. Our quantitative and qualitative findings indicated that ANC access, uptake and provision were affected during the pandemic. These three issues (ANC access, uptake, and provision) have been identified as a conduit that affects women attending hospitals for ANC and are essential for policymakers to develop strategies for increasing and sustaining ANC access, uptake, and provision during any future pandemic. Therefore, understanding ANC access, uptake, and provision is crucial for increasing the coverage and quality of ANC, especially in low and low-middle-income countries. 

8.Keywords

The author should select MESH terms for key word selection

Author response

Thank you for your comment, the revision has been made in line 65, page 3. 

Keywords: ANC access, uptake, provision, COVID-19, Ethiopia 

9. The first sentence needs citation–page 2, line49-50

Author response

Thank you for your comment, the revision has been made in line 68, page 3. 

1. Rothan HA, Byrareddy SN. The epidemiology and pathogenesis of coronavirus disease (COVID-19) outbreak. J Autoimmun. 2020; 109:102433.

10. Please avoid the use of reference repeatedly. E.g. reference 1, 3, etc. repeated more than two times. 

Author response

Thank you for your comment, the revision has been made in lines 69-78, page 3. 

11. Page 3 line 69, insert citation of reference (9) at the end of sentence. 

Author response

Thank you for your comment, the revision has been made in line 88, page 4. 

In 2016, the WHO recommended a minimum of eight ANC contacts for all pregnant women (11).

Methods

12. Please Rewrite as <Methods and Materials> 

Author response

Thank you for your comment, the revision has been made in line 109, page 4. 

Methods and materials

13. Setting

All information presented here are mentioned in the methods section. So please try to merge it or delete one them since in scientific paper writing form no need of redundancy. What are your study settings/institutions? It is not clearly stated, and try to focus on your study area and selected settings. 

Author response

Thank you for your comment, the revision has been made by merging, deleting, and focusing on our study setting. Described in detail in lines 111-122, page 5. 

This study is part of a larger mixed-methods study on the impact of COVID-19 on maternal and perinatal care at public hospitals in the Sidama region of southern Ethiopia. This mixed-methods study was carried out between 14 February 2022 and 10 May 2022 at fifteen public hospitals in the Sidama region. The Sidama region is the 10th newly established region in Ethiopia. The region is currently divided into five city administrations and 31 administrative divisions, known as 'Woredas'. In 2019/2020, the region's total population was 3,983,969, with 1,974,455 males and 2,009,514 females. This region has 928,265 women of reproductive age (15-49), 137,845 of whom gave birth in 2019/2020. This region comprises 14 primary hospitals, three general hospitals, one comprehensive specialised teaching hospital, 123 public health centres, 526 health posts and greater than 108 private clinics. Comprehensive emergency obstetric care (CEmOC) was available in 15 public hospitals in the Sidama region in 2020. 

14. Setting 

Please rewrite the whole paragraph of this section. No need of writing unnecessary information in study setting like the first sentence you wrote about study design <We implemented an interrupted time series (ITS) study design to estimate the average 126 changes in ANC provision during the first six months of the COVID-19 pandemic (March to 127 August 2020) at fifteen public hospitals in the Sidama region. ITS is a study design that 128 evaluates the impact of population-level interventions, including policy changes, infection 129 prevention programmes, and any pandemics like COVID-19 that have been implemented at a 130 clearly defined time (33)>. 

Author response

Thank you for your comment, the revision has been made in lines 132-138, pages 5 & 6. 

Data was collected from all fifteen public hospitals that provided CEmOC in the Sidama region. In this study, we included all pregnant women who uptake before COVID-19 (12 months from March 2019 to February 2020) and during COVID-19 (six months from March 2020 to August 2020) at public hospitals in the Sidama region. The first author and research assistants, who have MSc in clinical midwifery, extracted data from fifteen public hospitals HMIS in the Sidama region. 

15. Sample size not determined. Why? 

Author response

Thank you for your comment, the revision has been made in lines 132-136, page 5. 

This study included all pregnant women who uptake ANC before COVID-19 (12 months from March 2019 to February 2020) and during COVID-19 (six months from March 2020 to August 2020) at fifteen public hospitals in the Sidama region (the total number of women attending 15 hospitals for ANC from March 2019 to August 2020 was (N=21,000)).

16. Data processing and analysis

It lacks focus. It should describe about data entry and analysis, but not data collection methods, data collection procedures etc. 

Author response

Thank you for your comment, the revision has been made in lines 144-146, page 6. 

After screening the data, any questions related to data clarity were resolved by revisiting the hospitals and regional health HMIS offices. Data were imported from Microsoft Excel into STATA V.17 for analysis.

Qualitative methods

17. Please omit redundancy. Your method part is not clear as a whole. You have three section methods in this document (methods, quantitative methods, and Qualitative methods). You should have one comprehensive methods or a max of two sections of methods. 

Author response 

Thank you for your comment, the revision has been made in line 109, page 4 (Methods and materials), line 131, page 5 (data collection methods for quantitative strands), line 143, page 6 (data processing and analysis for quantitative strands), in line 172, page 7 (study approach for qualitative strands), in line 183, page 7,(study participant recruitment and sampling technique for qualitative strands), in line 193, page 7,(data collection tools and procedure for qualitative strands) and in line 207 page 8 (data processing and analysis for qualitative strands). 

18. Study participant recruitment and sampling technique

Author response 

Thank you for your comment, the study participant recruitment and sample technique are described in lines 184-192, page 7. 

We explained the purpose of the study to the hospital medical director, chief executive director and maternity care head to get permission to conduct the research. Two research assistants, who have MSc in clinical midwifery, explained in detail the purpose of the study for maternity care providers to volunteer to participate in the interview. We used a purposive sampling technique to recruit staff who provided maternity care before and during the pandemic. Participants gave written informed consent prior to being interviewed. We aimed to recruit approximately 20 participants (10 midwives and 10 obstetricians). Data were saturated at 24 interviews. We conducted another four interviews to confirm that data were saturated prior to ceasing qualitative data collection.

19. Who is your research assistant? Please clearly mentioned them 

Author response

Thank you for your comment, the revision has been made in line 137, page 5.

Research assistan

---

## [Editor Report · Decision Letter 1]

10 Aug 2023

PONE-D-23-07658R1

Impact of COVID-19 on antenatal care provision at public hospitals in Sidama region, Ethiopia: a mixed methods study

PLOS ONE

Dear Dr. Kassa,

Thank you for submitting your manuscript to PLOS ONE. After careful consideration, we have decided that your manuscript does not meet our criteria for publication and must therefore be rejected.

I am sorry that we cannot be more positive on this occasion, but hope that you appreciate the reasons for this decision.

Kind regards,

Ephraim Kumi Senkyire

Academic Editor

PLOS ONE

Additional Editor Comments:

Thank you for considering PLOSONE. Base on reviewer 2 detailed comments to refined your manuscript to meet the quality of mix-study, after careful cross-check with your response, most of the comment were not taken into consideration. e.g you claimed the study is a mixed method however, you failed to report or discuss the quantitative aspect of the study. this is one of the several vital comment you refused to address hence your manuscript can not be accepted in this journal.

- - - - -

---

## [Author Response · Author response to Decision Letter 1]

21 Oct 2023

Reviewer 2 comments Authors rebuttal 

1. The study is already over-researched, even a systematic review of meta-analysis has been done in Ethiopia entitled with <Impact of COVID-19 pandemic on utilisation of essential maternal healthcare services in Ethiopia: A systematic review and meta-analysis> or Link 

https://journals.plos.org/plosone/article?id=10.1371/journal.pone.0281260

So, what you added to your finding? 

 We appreciate the reviewer's concern regarding the over-researched issue related to our study. To the best of our search and knowledge, no study is exploring the impact of COVID-19 on ANC access, uptake, and provision in a rigorous method. Included studies in this systematic review and meta-analysis (the link you provided) and other studies used the impact of COVID-19 on ANC utilisation; however, they are quite different. The detailed difference between our study and other studies conducted in Ethiopia is as follows:

1. Most of the studies do not show ANC access, uptake, and provision before and during the pandemic. However, they showed ANC uptake during the pandemic without comparing it to before the pandemic, and they did not explore the impact of COVID-19 on ANC access, uptake, and provision. 

2. There are also methodological differences; the current study differs from previously published studies on the analysis model employed in the quantitative study and integration of quantitative and qualitative findings. Hence, previously published studies employed a binary logistic regression model, Paired t-test and non-parametric (Wilcoxon signed-rank) tests to estimate the trends of ANC uptake before and during the pandemic. However, the current study used interrupted time series analysis to estimate the trend of ANC uptake before and during the pandemic based on the recommendations from three biostatisticians. As described in the methods section of this paper, the rationale for using the interrupted time series analysis is found in lines 140-145 of page 5. Interrupted time series analysis is a robust statistical analysis method that recommends evaluating the impact of population-level interventions, including policy changes, infection prevention programmes, and pandemics like COVID-19 that have been experienced at a clearly defined time. 

3. As described in lines 152-155, page 5, we used the Poisson regression model because the outcome of interest is count data. Furthermore, in the current study, we controlled seasonal and other incidents' effects during the analysis. 

4. We used Braun and Clarke’s thematic analysis approach using NVivo 12 plus, which followed six phases to identify themes in our qualitative study. This has not been conducted in previously published articles from Ethiopia.

5. Our qualitative findings explored the impact of COVID-19 on ANC access, uptake, and provision in a rigorous method, while previous studies did not rigorously explore the impact of COVID-19 on ANC access, uptake, and provision independently.

6. We integrated the quantitative and qualitative findings in a joint display analysis based on the recommendations from John W. Creswell’s book (2014) and Fetters MD (2019), which showed three types of integration for concurrent mixed methods study data transformation, joint display technique and side-by-side (weaving) in the discussion section. In this study, we used the joint display approach, presented in Table 3; however, previous studies do not indicate which integration approach was used in concurrent mixed methods study.

7. In summary, the above justifications clearly show the difference between our current study from previously published works in the same area in Ethiopia. This study contributes to the body of evidence about critical issues and how stakeholders can maintain and enhance ANC access, uptake, and provision for future health systems in crisis related to the epidemic, pandemic, and any natural and man-made disaster. Additionally, this study adds to the body of knowledge and aims to contribute vital evidence to inform the planning and management of future population-level disasters (including epidemic/pandemic situations). 

2. The title should be rewrite in SMART form meaning that mainly it should clear where (specifically for local reader better to add < Sidam Region, Ethiopia> and when the study done. Thank you for your comment; the revision has been made in line 1, page 1. 

Impact of COVID-19 on antenatal care provision at public hospitals in Sidama region, Ethiopia: a mixed method study

Abstract

3. Methods section 

What is the exact data collection period? The revision has been made in lines 30-31, page 2. 

A concurrent mixed methods study was applied between 14 February 2022 and 10 May 2022 at fifteen public hospitals in the Sidama region.

4. What is your sample size? It is not clear to the reader. The following revision has been made in lines 25-28, pages 1 and 2:

The total numbers in the ANC1 cohort (at least one ANC contact) and ANC4 cohort (at least four ANC contacts) were 15,150 and 5,850, respectively, forming a combined final dataset of 21,000 women.

As we used data from all the women who accessed ANC from March 2019 to August 2020, there was no need for a sample size calculation. 

5. The author collect data before covid 19 (March 2019 to February 2020), so how to relate the study with covid 19? Our objective was to estimate the monthly attendance of women for ANC at public hospitals by comparing data before and data during the pandemic. This allowed us to assess monthly trends of ANC attendance, and to control for seasonal and other influencing effects. We extracted data from HMIS at 15 public hospitals for the monthly report before COVID-19 (March 2019 to February 2020) (twelve months of data) and during COVID-19 from March to August 2020 (six months of data) to estimate monthly trends of ANC uptake before and during the pandemic, as described in lines 38-35, page 2.

We used an interrupted time series model to estimate trends of monthly ANC uptake in the two periods (before and during COVID-19). Routinely collected monthly data were derived from the hospitals’ health management information system (HMIS) and imported into Stata version 17 for analysis.

6.Result

No information mentioned about the quantitative data in your result. So please include them. The revision has been made in lines 43-44, page 2. Our findings indicate that ANC1 uptake and ANC4 uptake, respectively, decreased by 0.7% and 1.8% in the first six months of the pandemic.

7.Conclusion

Authors should be selective to generalize the whole finding what they have gotten from the finding. It is not clear that why the authors select only three views (ANC access, uptake and provision) for conclusion? We acknowledge the reviewer’s concern, and ANC access, uptake, and provision were identified themes in the qualitative findings and pertinent findings in this study that add new insight to the impact of COVID-19 on ANC provision in this study. Our quantitative and qualitative findings indicated that ANC access, uptake and provision were affected during the pandemic. These three issues (ANC access, uptake, and provision) have been identified as a conduit that affects women attending hospitals for ANC and are essential for policymakers to develop strategies for increasing and sustaining ANC access, uptake, and provision during any future pandemic. Therefore, understanding ANC access, uptake, and provision is crucial for increasing the coverage and quality of ANC, especially in low and low-middle-income countries. 

8.Keywords

The author should select MESH terms for key word selection The revision has been made in line 58, page 3. 

Keywords: ANC access, uptake, provision, COVID-19, Ethiopia 

9. The first sentence needs citation–page 2, line49-50 The revision has been made in line 61, page 3. 

1. Rothan HA, Byrareddy SN. The epidemiology and pathogenesis of coronavirus disease (COVID-19) outbreak. J Autoimmun. 2020; 109:102433.

10. Please avoid the use of reference repeatedly. E.g. reference 1, 3, etc. repeated more than two times. The revision has been made in lines 64-71, page 3. 

11. Page 3 line 69, insert citation of reference (9) at the end of sentence. The revision has been made in line 81, page 3. 

In 2016, the WHO recommended a minimum of eight ANC contacts for all pregnant women.

Methods

12. Please Rewrite as <Methods and Materials> We appreciate the reviewer's concerns; we did not use any materials. Therefore, we put it as “methods.” 

13. Setting

All information presented here are mentioned in the methods section. So please try to merge it or delete one them since in scientific paper writing form no need of redundancy. What are your study settings/institutions? It is not clearly stated, and try to focus on your study area and selected settings. The revision has been made by merging, deleting, and focusing on our study setting. Described in detail in lines 103-114, page 4. 

The revision has been made as follows:

This study is part of a larger mixed-methods investigation of the impact of COVID-19 on maternal and perinatal care at 15 public hospitals in the Sidama region of southern Ethiopia, carried out between 14 February and 10 May 2022. Sidama is the 10th newly established region in Ethiopia. The region is currently divided into 5 city administrations and 31 administrative divisions, known as ‘Woredas’. In 2019/2020, the region’s total population was 3,983,969, with 1,974,455 males and 2,009,514 females. Sidama has 928,265 women of reproductive age (15–49), 137,845 of whom gave birth in 2019/2020. Regarding healthcare provision, the region has 14 primary hospitals, 3 general hospitals, 1 comprehensive specialised teaching hospital, 123 public health centres, 526 health posts, and over 108 private clinics. Comprehensive emergency obstetric care was available in 15 public hospitals in Sidama during 2020.

14. Setting 

Please rewrite the whole paragraph of this section. No need of writing unnecessary information in study setting like the first sentence you wrote about study design <We implemented an interrupted time series (ITS) study design to estimate the average 126 changes in ANC provision during the first six months of the COVID-19 pandemic (March to 127 August 2020) at fifteen public hospitals in the Sidama region. ITS is a study design that 128 evaluates the impact of population-level interventions, including policy changes, infection 129 prevention programmes, and any pandemics like COVID-19 that have been implemented at a 130 clearly defined time (33)>. The revision has been made as follows:

Data were collected from all 15 public hospitals providing comprehensive emergency obstetric care in Sidama at the time of the study. The sample included all pregnant women who attended ANC in the 12 months before the COVID-19 pandemic (March 2019 to February 2020) and during the six months of the pandemic (March to August 2020), totalling 21,000.

15. Sample size not determined. Why? The following revision has been made in lines 125-131, page 5:

Data were collected from all 15 public hospitals providing comprehensive emergency obstetric care in Sidama at the time of the study. The sample included all pregnant women who attended ANC in the 12 months before the COVID-19 pandemic (March 2019 to February 2020) and during the six months of the pandemic (March to August 2020), totalling 21,000 women. As we used data from all the women who attended ANC from March 2019 to August 2020, there was no need for a sample size calculation. 

16. Data processing and analysis

It lacks focus. It should describe about data entry and analysis, but not data collection methods, data collection procedures etc. The revision has been made in lines 140-144, page 5. 

After screening the data, any questions regarding data clarity were resolved by revisiting the hospitals and regional HMIS offices. Data were imported from Microsoft Excel into Stata version 17 for analysis. We performed an interrupted time series analysis (ITSA) to estimate trends in the uptake of ANC across two periods: before COVID-19 (March 2019 to February 2020) and during COVID-19 (March to August 2020).

Qualitative methods

17. Please omit redundancy. Your method part is not clear as a whole. You have three section methods in this document (methods, quantitative methods, and Qualitative methods). You should have one comprehensive methods or a max of two sections of methods. The revision has been made, methods in one section in line 101, page 4 

Methods 

18. Study participant recruitment and sampling technique The study participant recruitment and sample technique are described in lines 188-198, page 7. 

Within each chosen hospital, we explained the study’s purpose to the hospital medical director, chief executive director, and maternity care head, seeking their permission to conduct the research. Subsequently, two research assistants (both with an MSc in clinical midwifery) explained the study’s purpose in detail to maternity care providers who volunteered to be interviewed. We used purposive sampling to recruit staff who provided maternity care both before and during the pandemic. All participants provided written informed consent prior to being interviewed. We aimed to recruit approximately 20 participants (10 midwives and 10 obstetricians). Data reached saturation at 24 interviews. We conducted another four interviews to confirm that data were saturated before ending qualitative data collection.

19. Who is your research assistant? Please clearly mentioned them Research assistants who have MSc in clinical midwifery were not included as authors in this manuscript. As they were data collectors, they did not meet the criteria for co-authorship of the paper. The responsibility of research assistants was to facilitate the data collection process, for example, in the study participants’ selection process and facilitate the interviews.

“We” is used in this manuscript for those who are included as authors.

20. What are your study participants? Patient or healthcare provider As stated in lines 176-177 on pages 6 & 7, the study participants are maternity care providers (midwives, obstetrics and gynaecology residents, integrated emergency surgical officers and obstetricians).

21. Why the authors used a purposive sampling technique? Is that good to use non probable sampling methods to write scientific papers important for scientific evidence? In my view it is not acceptable. Purposive sampling was used for the qualitative component of this mixed methods study, as per usual practice. We prepared open-ended questions that addressed our objectives, and it was piloted outside of the study area. It is common in qualitative research to use a purposive sampling technique as it is necessary to be sure the participants have the appropriate experience to contribute to the data. The qualitative study focuses on the depth of individual views and experiences related to our study objectives; therefore, the participants need to have worked in the relevant hospital settings before and during the pandemic. The last author on this paper is a senior academic with a PhD and more than a decade’s experience in qualitative methodologies and methods.

22. Please insert this sentence in <Participants provided written informed consent prior to being interviewed> in Ethical approval and consent section. The revision has been made in lines 234, page 8.

Participants provided written informed consent prior to being interviewed.

23. What about Study participant recruitment and sampling technique of quantitative methods?? The following revision has been made in lines 125-131, page 5. 

Data were collected from all 15 public hospitals providing comprehensive emergency obstetric care in the Sidama region at the time of the study. The sample included all pregnant women who attended ANC in the 12 months before the COVID-19 pandemic (March 2019 to February 2020) and during the six months of the pandemic (March to August 2020), totalling 21,000 women.

We collected data from the medical records of all women who attended ANC from March 2019 to August 2020, therefore, there was no need for recruitment of individuals or for a sample size calculation. In addition, interrupted time serie

---

## [Decision Letter · Decision Letter 2]

24 Jan 2024

PONE-D-23-07658R2Impact of COVID-19 on antenatal care provision at public hospitals in Sidama region, Ethiopia: a mixed methods studyPLOS ONE

Dear Dr. Kassa,

Thank you for submitting your manuscript to PLOS ONE. After careful consideration, we feel that it has merit but does not fully meet PLOS ONE’s publication criteria as it currently stands. Therefore, we invite you to submit a revised version of the manuscript that addresses the points raised during the review process. Please submit your revised manuscript by 27 Feb 2024. If you will need more time than this to complete your revisions, please reply to this message or contact the journal office at plosone@plos.org. Please include the following items when submitting your revised manuscript:A rebuttal letter that responds to each point raised by the academic editor and reviewer(s). You should upload this letter as a separate file labeled 'Response to Reviewers'.A marked-up copy of your manuscript that highlights changes made to the original version. You should upload this as a separate file labeled 'Revised Manuscript with Track Changes'.An unmarked version of your revised paper without tracked changes. You should upload this as a separate file labeled 'Manuscript'.

We look forward to receiving your revised manuscript.

Kind regards,

Fekede Asefa Kumsa, PhD

Academic Editor

PLOS ONE

Journal Requirements:

Additional Editor Comments (if provided):

One of the significant issues in this manuscript is a lack of focus and idea fragmentation. In the findings section, similar ideas and quotes appear in multiple places. For instance, the community's perception of health facilities as the epicenter of COVID-19 infection is displayed at various points under different themes or sub-themes. The same issue arises with transportation-related issues. There is a misalignment between some themes/subthemes and the provided descriptions. For example, in the sub-theme 'Community discrimination against women attending hospital,' the description addresses both community discrimination against women and care providers while the subtheme is about women. It is crucial for the theme/subtheme and the description to align cohesively. The finding section benefit from proper resynthesis. Additionally, the manuscript focuses on the impact of COVID-19 on antenatal care provision. However, unrelated issues regarding vaccine hesitancy are introduced in the findings section without proper context.

Reviewers' comments:

Reviewer's Responses to Questions

**Comments to the Author**

1. If the authors have adequately addressed your comments raised in a previous round of review and you feel that this manuscript is now acceptable for publication, you may indicate that here to bypass the “Comments to the Author” section, enter your conflict of interest statement in the “Confidential to Editor” section, and submit your "Accept" recommendation.

Reviewer #3: (No Response)

Reviewer #4: (No Response)

2. Is the manuscript technically sound, and do the data support the conclusions?

Reviewer #3: No

Reviewer #4: Partly

3. Has the statistical analysis been performed appropriately and rigorously? 

Reviewer #3: I Don't Know

Reviewer #4: Yes

4. Have the authors made all data underlying the findings in their manuscript fully available?

Reviewer #3: No

Reviewer #4: Yes

5. Is the manuscript presented in an intelligible fashion and written in standard English?

Reviewer #3: No

Reviewer #4: Yes

6. Review Comments to the Author

Reviewer #3: Reviewer comments

Manuscript Number: PONE-D-23-07658R2

Full Title: Impact of COVID-19 on antenatal care provision at public hospitals in Sidama region, Ethiopia: a mixed methods study

Generally, the manuscript need thorough English language edition. The punctuation and grammar need correction to facilitate easy understanding for readers.

Title: Is effect or impact appropriate term to indicate results of covid-19 during mentioned times (six months, March to August 2020). This times as authors may remember was when there were complete to partial closure of services (the government declared a five-month national state of emergency starting April 8th, 2020), it was not time when services put in to place or cease of closure.

Abstract

Background: what was study gaps that authors want to uncover?

Method: Did authors calculate minimum sample size? How authors selected these facilities and participants? How authors maintained data quality? How quantitative data were reported?

Result: line 37 and 38, the sentence is not clear. Was the effect of covid-19 significant? Line 41 subtheme ‘COVID-19 vaccine hesitancy’ was identified by authors. Was there a vaccine during specified time? I recommend these subthemes categorized under their themes, so that effect of covid-19 seen boldly.

Conclusion and recommendation: is qualitative or quantitative result support this conclusion? Or both? Authors did not indicate adequate finding that support their conclusion in result part of the abstract. Are your recommendation supported by your evidence and updated? What is the study implication?

Introduction

Authors did not indicate study gaps, why this study is needed. What is the significance of this study? Why qualitative study needed? What is the objective of qualitative study?

Method

Line 100 what specific design was used for quantitative study? How about qualitative study? These two terms are broad terms to describe design, authors need to explain specific design they employed for both type of study.

Line 107, rewrite the subtitle. ‘Data collection method for quantitative data’ The content should also clearly indicate the tool used, who collected and how you collected the data.

Collection tools and procedure for qualitative data

How authors interviewed those who have difficulties in speaking official language?

Authors failed to explain how they assured quality of data for both quantitative and qualitative methods.

Result

How many was the response rate? Was there any missing data? How it was managed?

Change the following subtitle ‘Qualitative results’ to appropriate subtopic. Check its content and choose.

The authors are not expected to write ‘quote’ for each paragraphs. Too many quote indicate poor analysis of the data.

Discussion

Authors should focus on the major findings. This discussion is not easy to understand and focused.

Conclusion

Is the disruption is still there? Authors should cautiously conclude and recommend based on their findings. It may contribute to knowledge of what happened during covid-19. Otherwise, I don’t think this conditions are still there in Sidama.

Reviewer #4: (No Response)

7. PLOS authors have the option to publish the peer review history of their article (what does this mean?). If published, this will include your full peer review and any attached files.

Reviewer #3: No

Reviewer #4: No

---

## [Author Response · Author response to Decision Letter 2]

19 Feb 2024

Dear Editor and reviewers, 

Thank you for taking the time to review our manuscript and for your constructive comments and suggestions, which are vital to improving the quality of the manuscript. We addressed your comments point by point, provided clarifications, and incorporated your suggestion into the manuscript as follows:

Editor comments 

1. In the findings section, similar ideas and quotes appear in multiple places. For instance, the community's perception of health facilities as the epicenter of COVID-19 infection is displayed at various points under different themes or sub-themes. 

Authors response

Thank you for your comment; the following revision has been made in lines 335 and 337, page 15. The term ‘Epicentre of COVID-19 infection’ only appears in the ‘Fear of contracting COVID-19’ subtheme. 

2. The same issue arises with transportation-related issues.

Authors response 

Thank you for your comment; the following revision has been made in lines 271 and 275, page 13. The term ‘transport’ only appears in the ‘Shortage of resources’ subtheme. 

3. There is a misalignment between some themes/subthemes and the provided descriptions. For example, in the sub-theme 'Community discrimination against women attending hospital,' the description addresses both community discrimination against women and care providers while the subtheme is about women. 

Authors response

The following revision has been made in line 288, page 13.

Community discrimination against those attending the hospital

4. The finding section benefit from proper resynthesis. Additionally, the manuscript focuses on the impact of COVID-19 on antenatal care provision. However, unrelated issues regarding vaccine hesitancy are introduced in the findings section without proper context.

Authors response

Dear editor, we appreciate your concern regarding the relation between ANC and vaccine hesitancy. 

Vaccine hesitancy could affect ANC provision, as stated in Table 3.

The qualitative findings demonstrate that COVID-19 vaccine hesitancy impacted ANC provision. The vaccine hesitancy amongst maternity care providers could have increased the reluctance of pregnant women to be vaccinated, since the providers may not have persuaded them of the vaccine’s benefits during pregnancy. Furthermore, vaccine-hesitant maternity care providers’ fear of contracting the virus when providing ANC could have further reduced ANC provision. 

5. The finding section benefit from proper resynthesis. 

Authors response

Thank you for your comment; the qualitative finding synthesis was reviewed iteratively as per the approach of Clarke and Braun in thematic analysis, and modifications have been made across the synthesis of the findings. 

Reviewer 3 comments 

1. Generally, the manuscript need thorough English language edition. The punctuation and grammar need correction to facilitate easy understanding for readers.

Authors response

Thank you for your comment; the manuscript has been reviewed in detail by two senior authors, both native English speakers and an English expert person. Grammarly Premium was used to check the manuscript's spelling, grammar, and punctuation. Additionally, the manuscript was edited and proofread by Elsevier language editing services.

2. Is effect or impact appropriate term to indicate results of covid-19 during mentioned times (six months, March to August 2020). 

Authors response

We appreciate your concern; we believe ‘impact’ is an appropriate term for this study. Impact is often used to describe the immediate influence of COVID-19 on ANC.

3. what was study gaps that authors want to uncover? 

Authors response

The following revision has been made in lines 18 and 19, page 1:

There is a paucity of studies on the impact of COVID-19 on antenatal care access, uptake, and provision in Ethiopia.

4. Did authors calculate minimum sample size? 

Authors response

As stated in lines 131-135, page 5, data were collected from all 15 public hospitals providing comprehensive emergency obstetric care in the Sidama region at the time of the study. The sample included all pregnant women who attended ANC in the 18 months before the COVID-19 pandemic (March 2019 to February 2020) and during the six months of the pandemic (March to August 2020), totalling 21,000 women. As we used data from all the women who attended ANC from March 2019 to August 2020, there was no need for a sample size calculation. 

Furthermore, this study is part of a broader mixed methods study, and a minimum sample size was calculated. In fact, it is mandatory to calculate the sample size, so the single population proportion formula was used. The health coverage in the Sidama region 2019/2020 was 77.2%. The level of significance was 5 %(a=0.05), the margin of error was 3 % (d=0.03), and the non-responsive rate was 10%; the final sample was 826.

5. How authors selected these facilities? 

Authors response

All 15 public hospitals that provided comprehensive emergency obstetrics in the Sidama region were included in the quantitative study. However, as stated in lines 172 -180, pages 6 and 7, four hospitals were selected for the qualitative study. These four public hospitals (including two primary hospitals, one general hospital and one specialised hospital) were chosen for the qualitative study based on the caseload maternity care services provided and the order in which COVID-19 cases were initially reported in the Sidama region. Three different types of hospitals were selected: primary, general, and one specialised hospital that served as a referral centre for the Sidama region and the surrounding population in the Oromia region. This selection allowed for a nuanced understanding of the impact of the pandemic on various tiers of hospitals and their preparedness, response efficiency and the challenges they faced. 

6. How authors selected these participants? 

Authors response

The quantitative data extraction was stated in lines 132-135, page 5:

Data were collected from all 15 public hospitals providing comprehensive emergency obstetric care in the Sidama region at the time of the study. The sample included all pregnant women who attended ANC in the 12 months before the COVID-19 pandemic (March 2019 to February 2020) and during the six months of the pandemic (March to August 2020), totalling 21,000.

For qualitative study, the study participants' selection was described in lines 182 -190, page 7. 

Within each chosen hospital, we explained the study’s purpose to the hospital medical director, chief executive director, and maternity care head, seeking their permission to conduct the research. Subsequently, two research assistants (both with an MSc in clinical midwifery) explained the study’s purpose in detail to maternity care providers who volunteered to be interviewed. We used purposive sampling to recruit staff who provided maternity care both before and during the pandemic. All participants provided written informed consent prior to being interviewed. We aimed to recruit approximately 20 participants (10 midwives and 10 obstetricians). Data reached saturation at 24 interviews. We conducted another four interviews to confirm that data were saturated before ending qualitative data collection.

7. How authors maintained data quality? 

Authors response

Thank you for your comment; the quantitative data quality was maintained by revisiting the hospitals and HMIS offices to address any missing data in the provided Excel spreadsheet, as stated in lines 142 and 143, page 5. Data cleaning and preprocessing were done to check missing data. In addition, based on the study objective, the authors selected robust methods of analysis, which is an interrupted time series analysis through a poison regression model. 

The interview guide was approved by the Ethics Committees at UTS HREC and IRB Hawassa University.

For the qualitative study, the data quality was maintained by preparing a study guide and conducting a pilot study, as stated in line 197, page 7:

The audio recordings were transcribed immediately and listened to iteratively. Simultaneously, bilingual researchers transcribed and translated transcripts into English to check consistency. The transcriptions were imported into NVivo software (QSR International, version 12 Plus) to manage the overall data analysis, as stated in lines 206-209, page 8. All authors reviewed the themes and subthemes in the thematic analysis phases (from coding to writing a report) and approved the final themes, as stated in lines 213 -215, page 8.

8. How quantitative data were reported? 

Authors response

The quantitative data were reported based on assumptions of the model (Poisson regression) using a P value less than 0.05 (p<0.05), which is considered statistically significant, and using a 95% confidence interval. The incidence rate ratio (IRR) of ANC1 and ANC4 is reported in the figure, table, and texts, which are stated in lines 150-157, page 6. In addition, we based our reporting on journal submission guidelines.

The mean monthly incidence rate ratio (IRR) of ANC uptake was calculated with a 95% confidence interval (CI) using a Poisson regression model (34, 35) with pre-COVID-19 data as the reference. A Poisson regression model was suitable because the monthly reports of ANC provision comprised count data (non-negative integer values). In ITSA, a Poisson regression model (36) performs better than an autoregressive integrated moving average (ARIMA) model, which is more conventionally used for real-valued time series data. Differences are considered statistically significant at a p-value of less than 0.05 (p < 0.05).

9. Result: line 37 and 38, the sentence is not clear. Was the effect of covid-19 significant? 

Authors response

Thank you for your comment: the revision has been made in lines 40 and 41, page 2:

Our findings indicate a significant monthly decrease of 0.7% in ANC1 and 1.8% in ANC4 during the first six months of the pandemic.

10. Line 41 subtheme ‘COVID-19 vaccine hesitancy’ was identified by authors. Was there a vaccine during specified time? I recommend these subthemes categorized under their themes, so that effect of covid-19 seen boldly.

Authors response 

We understand your concern: the data were collected between 14 February and 10 May 2022. During the data collection, COVID-19 vaccinations were available. Some study participants reported that they did not get vaccinations, for a variety of reasons. These factors could potentially have disrupted the provision of ANC. Further details are provided in Table 3.

The qualitative findings demonstrate that COVID-19 vaccine hesitancy impacted ANC provision. Such hesitancy in maternity care providers could have increased the reluctance of pregnant women to be vaccinated, since the providers would not attempt to persuade them of the vaccine’s benefits during pregnancy. Furthermore, vaccine-hesitant maternity care providers’ fear of contracting the virus when providing ANC could have further reduced ANC provision.

11. Conclusion and recommendation: is qualitative or quantitative result support this conclusion? Or both? Authors did not indicate adequate finding that support their conclusion in result part of the abstract. Are your recommendation supported by your evidence and updated? What is the study implication? 

Authors response

Thank you for your comment; the following revision has been made in lines 51-57, pages 2 and 3, and the conclusion and recommendation are derived from our concurrent mixed methods findings. 

A lack of medical supplies, discrimination against those attending the hospital, and the absence of ANC guidelines for care provision led to disrupted access, uptake, and provision of ANC during COVID-19. To mitigate disrupted ANC access, uptake and provision, ANC clinics should be equipped with medical supplies. It is crucial to maintain rapport between the community and maternity care providers and provide training for maternity care providers regarding the adapted/adopted guidelines during COVID-19 at the hospital grassroots level for use in the current and future pandemics.

12. Authors did not indicate study gaps, why this study is needed. What is the significance of this study? Authors response

Thank you for your comment; the revision has been made in lines 97-109, page 4. 

In the early stage of the COVID-19 pandemic, Ethiopian government and NGOs shifted their focus towards containing the spread of the virus by implementing a range of measures. These measures included declaring a state of emergency, reducing the passenger capacity in public transport by half, imposing a lockdown and encouraging people to stay at home (27). Consequently, the lockdown measures resulted in job losses for many women (28). These factors posed significant challenges to women’s ability to meet their basic needs (28). As a result, the provision of ANC has been and continues to be, impacted by the direct and indirect consequences of COVID-19 (29). However, the existing studies in Ethiopia on the impact of the COVID-19 pandemic have not rigorously explored its impact on maternity care services, specifically ANC access, uptake, and provision. The paucity of studies on the impact of COVID-19 on ANC access, uptake and provision in Ethiopia made it essential to conduct this study to estimate and explore the impact of COVID-19 on the country’s ANC access, uptake, and provision. 

In addition, most of the studies show in their titles that they have conducted a mixed methods study, but in many cases, it was simply qualitative and quantitative studies. When studies lack integration and do not tell us which techniques of integration were used and applied in the method, result, or discussion section, this is not truly mixed methods. However, in this study, we integrated the quantitative and qualitative findings in a joint display technique based on the recommendations from John W. Creswell’s book (2014) and Fetters MD (2019), which showed three types of integration for concurrent mixed methods- study data transformation, joint display technique and side-by-side (weaving) in the discussion section. In this study, we used the joint display approach, presented in Table 3; however, previous studies do not indicate which integration approach was used.

13. Why qualitative study needed? What is the objective of qualitative study?

Authors response

Dear reviewer, we appreciate your concern regarding the study methods. We applied a concurrent mixed methods study to estimate and explore the impact of COVID-19 on ANC at public hospitals. The quantitative findings estimate the trends of ANC in the first 12 months before COVID-19 and in the first six months of the pandemic. Simultaneously, the qualitative findings explore maternity care providers' experiences and perceptions of ANC provision before and during the pandemic. The objective of this mixed methods study was to explore whether the qualitative findings confirmed or disconfirmed the quantitative findings. 

14. Line 100 what specific design was used for quantitative study? 

Authors response 

Thank you for your comment; the revision has been made in lines 124, page 5 

An interrupted time series design was applied for the quantitative component of the study.

15. How about a qualitative study? 

Authors response

As stated in line 169, page 6, we adopted an exploratory design to investigate maternity care providers’ views and experiences of the impact of COVID-19 on ANC provision in the Sidama region.

16. These two terms are broad terms to describe design, authors need to explain specific design they employed for both type of study. 

Authors response

As stated in line 122, page 5, in this study we used a concurrent mixed-methods design. 

17. Line 107, rewrite the subtitle. ‘Data collection method for quantitative data’ The content should also clearly indicate the tool used, who collected and how you collected the data.

Authors response

Thank you for your comment; the revision has been made in line 131, page 5: 

Data collection methods for quantitative data

This study is part of a broader mixed methods study, focusing on the outcome variables ANC1, ANC4, and essential medicines. These variables were extracted from the HMIS office in each hospital. The data was obtained from the HMIS office at

---

## [Editor Report · Decision Letter 3]

8 Mar 2024

PONE-D-23-07658R3Impact of COVID-19 on antenatal care provision at public hospitals in Sidama region, Ethiopia: a mixed methods studyPLOS ONE

Dear Dr. Kassa,

Thank you for submitting your manuscript to PLOS ONE. After careful consideration, we feel that it has merit but does not fully meet PLOS ONE’s publication criteria as it currently stands. Therefore, we invite you to submit a revised version of the manuscript that addresses the points raised during the review process. Please submit your revised manuscript by Apr 22 2024 11:59PM. If you will need more time than this to complete your revisions, please reply to this message or contact the journal office at plosone@plos.org. Please include the following items when submitting your revised manuscript:A rebuttal letter that responds to each point raised by the academic editor and reviewer(s). You should upload this letter as a separate file labeled 'Response to Reviewers'.A marked-up copy of your manuscript that highlights changes made to the original version. You should upload this as a separate file labeled 'Revised Manuscript with Track Changes'.An unmarked version of your revised paper without tracked changes. You should upload this as a separate file labeled 'Manuscript'.If applicable, we recommend that you deposit your laboratory protocols in protocols.io to enhance the reproducibility of your results. Protocols.io assigns your protocol its own identifier (DOI) so that it can be cited independently in the future. For instructions see: https://journals.plos.org/plosone/s/submission-guidelines#loc-laboratory-protocols. Additionally, PLOS ONE offers an option for publishing peer-reviewed Lab Protocol articles, which describe protocols hosted on protocols.io. Read more information on sharing protocols at https://plos.org/protocols?utm_medium=editorial-email&utm_source=authorletters&utm_campaign=protocols.

We look forward to receiving your revised manuscript.

Kind regards,

Fekede Asefa Kumsa, PhD

Academic Editor

PLOS ONE

Journal Requirements:

Additional Editor Comments:

Dear authors,

Thank you for addressing most of the the reviewers comments. There still some minor issues need to be addressed.

1. I would suggest the authors work on the abstract section to make it more appealing, especially the result section. The second sentence of the conclusion and recommendation section seems better fit into the result section and please move it up. Please don't introduce an abbreviation in the abstract if you are not gong to use it in this section. e.g., HMIS. Your abbreviation use is actually inconsistent throughout the document. For some, you provided multiple full text description (e.g., WHO), while for some others, you started using it without providing a full text discerption at the first use (e.g., ANC). Please make sure your abbreviations use is consistent and follows the standard.

2. In the abstract section, you stated that you collected 12 months' data before the occurrence of COVID-19, while in the method section it says 18 months data were collected. Please make sure you are consistent. In addition, please also discuss the appropriateness of comparing 12 or 18 months data (before COVID-19) with 6 months data (during COVID-19).

3. Please move table 2 to supplementary file.

4. Under the community discrimination against those attending the hospital sub-theme, you clearly discussed about the community discrimination against those attending the hospital. However, you gave little emphases on its link to the access to ANC. You have tried to say a few things in a second sentence of this section, but didn't substantiate it with the data from the participants. Please clearly show the link of this sub-theme to the access to ANC by substantiating it with the data.

5. Neither your discerption nor the provided quate still doesn't show the influence of COVID-19 vaccine hesitancy on ANC service utilization. You have tried to respond on the response letter, but the description should appear on the manuscript too. Please make sure that this claim is substantiated with evidence came from the participants. It shouldn't be the authors speculation.

Best regards,

---

## [Author Response · Author response to Decision Letter 3]

12 Mar 2024

Dear Editor, 

Thank you for taking the time to review our manuscript and for your constructive comments and suggestions, which are vital to improving the quality of the manuscript. We addressed your comments point by point, provided clarifications, and incorporated your suggestion into the manuscript as follows:

Editor comments 

1. The second sentence of the conclusion and recommendation section seems better fit into the result section and please move it up. 

Authors response

Thank you for your comment; the following revision has been made in lines 40-44, page 2, as follows:

A lack of medical supplies, fear of contracting COVID-19, inadequate personal protective equipment, discrimination against those attending the hospital, and the absence of antenatal care guidelines for care provision, COVID-19 vaccine hesitancy and long waiting times for ANC led to disrupted access, uptake, and provision of antenatal care during COVID-19.

2. Please don't introduce an abbreviation in the abstract if you are not gong to use it in this section. e.g., HMIS. Your abbreviation use is actually inconsistent throughout the document. For some, you provided multiple full text description (e.g., WHO), while for some others, you started using it without providing a full text discerption at the first use (e.g., ANC). Please make sure your abbreviations use is consistent and follows the standard. 

Authors response

Thank you for your comment; the revision has been made. The abstract has been written without any abbreviation, adhering to the prescribed wordage.

The revision has been made in the introduction section, line 77, page 3 as follows: 

antenatal care (ANC)

3. In the abstract section, you stated that you collected 12 months' data before the occurrence of COVID-19, while in the method section it says 18 months data were collected. Please make sure you are consistent. In addition, please also discuss the appropriateness of comparing 12 or 18 months data (before COVID-19) with 6 months data (during COVID-19). 

 Authors response

Thank you for your comment; the revision has been made in line 135, page 5.

Twelve months before the COVID-19 pandemic (March 2019 to February 2020).

 In addition, we aimed to assess the ANC uptake during the same period initially (March to August 2019 before COVID-19) and March to August 2020 during COVID-19 to control seasonal effects. After consulting a biostatistician, we decided to include the period from September 2019 to February 2020, whether ANC has been affected by other incidences or not, to control other incidents effects as stated in lines 140-142, page 5. This study aims to assess any monthly changes in ANC1 and ANC4 attendance before COVID-19 (March 2019 to February 2020) and during COVID-19 (March to August 2020). The study was conducted while controlling seasonal effects and other incidents impacting ANC1 and ANC4 attendance. Therefore, this study compared monthly changes in ANC1 and ANC4 attendance before COVID-19 (March 2019 to February 2020) and during COVID-19 (March to August 2020). 

4. Please move table 2 to supplementary file. 

Authors response

The revision has been made by moving Table 2 to Supplementary Table 2. 

5. Under the community discrimination against those attending the hospital sub-theme, you clearly discussed about the community discrimination against those attending the hospital. However, you gave little emphases on its link to the access to ANC. You have tried to say a few things in a second sentence of this section, but didn't substantiate it with the data from the participants. Please clearly show the link of this sub-theme to the access to ANC by substantiating it with the data. 

 Authors response

Thank you for your comment; the following revision has been made in lines 306 and 317, page 14, as follows:

resulting in limited access to ANC. 

Consequently, this discrimination resulted in limited access to ANC. 

6. Neither your discerption nor the provided quate still doesn't show the influence of COVID-19 vaccine hesitancy on ANC service utilization. You have tried to respond on the response letter, but the description should appear on the manuscript too. Please make sure that this claim is substantiated with evidence came from the participants. It shouldn't be the authors speculation. 

 Authors response

Thank you for your comment; the following revision has been made in lines 432-434, page 18, as follows:

If maternity care providers are hesitant about the COVID-19 vaccine, it may impact pregnant women's confidence in the vaccine's safety and efficacy. This could lead to lower vaccine rates and hinder ANC provision. In addition, at the beginning of the COVID-19 vaccine rollout, priority was given to frontline healthcare workers to enhance medical care, including ANC provision.

---

## [Editor Report · Decision Letter 4]

26 Mar 2024

Impact of COVID-19 on antenatal care provision at public hospitals in Sidama region, Ethiopia: a mixed methods study

PONE-D-23-07658R4

Dear Dr. Kassa,

We’re pleased to inform you that your manuscript has been judged scientifically suitable for publication and will be formally accepted for publication once it meets all outstanding technical requirements.

Kind regards,

Fekede Asefa Kumsa, PhD

Academic Editor

PLOS ONE

---

## [Editor Report · Acceptance letter]

8 Apr 2024

PONE-D-23-07658R4 

PLOS ONE

Dear Dr. Kassa, 

I'm pleased to inform you that your manuscript has been deemed suitable for publication in PLOS ONE. Congratulations! Your manuscript is now being handed over to our production team.

Kind regards, 

on behalf of

Dr. Fekede Asefa Kumsa 

Academic Editor

PLOS ONE